# Contextualizing genetic risk score for disease screening and rare variant discovery

Dan Zhou [1], Dongmei Yu[2,3], Jeremiah M. Scharf[2,3,4,5], Carol A. Mathews[6], Lauren McGrath[7], Edwin Cook[8], S. Hong Lee[9,10], Lea K. Davis [1,11,12 ✉] & Eric R. Gamazon [1,13,14 ✉]

Studies of the genetic basis of complex traits have demonstrated a substantial role for common, small-effect variant polygenic burden (PB) as well as large-effect variants (LEV, primarily rare). We identify sufficient conditions in which GWAS-derived PB may be used for well-powered rare pathogenic variant discovery or as a sample prioritization tool for whole-genome or exome sequencing. Through extensive simulations of genetic architectures and generative models of disease liability with parameters informed by empirical data, we quantify the power to detect, among cases, a lower PB in LEV carriers than in non-carriers. Furthermore, we uncover clinically useful conditions wherein the risk derived from the PB is comparable to the LEV-derived risk. The resulting summary-statistics-based methodology (with publicly available software, PB-LEV-SCAN) makes predictions on PB-based LEV screening for 36 complex traits, which we confirm in several disease datasets with available LEV information in the UK Biobank, with important implications on clinical decision-making.

[1] Vanderbilt Genetics Institute, Division of Genetic Medicine, Department of Medicine, Vanderbilt University Medical Center, Nashville, TN, USA. [2] Psychiatric and Neurodevelopmental Genetics Unit, Center for Genomic Medicine, Department of Psychiatry, Massachusetts General Hospital, Boston, MA, USA. [3] Stanley Center for Psychiatric Research, Broad Institute of MIT and Harvard, Cambridge, MA, USA. [4] Department of Neurology, Massachusetts General Hospital, Boston, MA, USA. [5] Department of Neurology, Brigham and Women's Hospital, Boston, MA, USA. [6] Department of Psychiatry, Genetics Institute, University of Florida, Gainesville, FL, USA. [7] Department of Psychology, University of Denver, Denver, CO, USA. [8] Department of Psychiatry, Institute for Juvenile Research, University of Illinois at Chicago, Chicago, IL, USA. [9] Australian Centre for Precision Health, University of South Australia Cancer Research Institute, University of South Australia, Adelaide, SA, Australia. [10] UniSA Allied Health and Human Performance, University of South Australia, Adelaide, SA, Australia. [11] Department of Psychiatry and Behavioral Sciences, Vanderbilt University Medical Center, Nashville, TN, USA. [12] Department of Biomedical Informatics, Vanderbilt University Medical Center, Nashville, TN, USA. [13] Clare Hall, University of Cambridge, Cambridge, United Kingdom. [14] MRC Epidemiology Unit, University of Cambridge, Cambridge, United Kingdom. ✉email: lea.k.davis@vumc.org; ericgamazon@gmail.com

Over the past several years, human genetics has shifted from controversies over the relative importance of rare or common variation in the genetic architecture of complex disease[1–9], towards realization of the need for integrative analyses of variation across the entire frequency spectrum[10]. While the individual effects of common single nucleotide variants (SNVs) on complex traits are often modest, a substantial proportion of complex trait heritability can be explained by common variants considered in aggregate[1–5,11,12]. In contrast, large-effect variants (LEVs) are typically rare in the population but confer substantial genetic risk for disease. For example, rare and de novo loss of function (LOF) single nucleotide variants and large, rare, genic copy number variants (lrgCNVs) substantially increase risk for many common conditions including neuropsychiatric, endocrine, and cancer phenotypes[6–8,13,14]. Although these LEVs confer substantial risk, they also exhibit incomplete penetrance, where penetrance is defined as the probability of developing the disease given the genotype. Indeed, the reported penetrance of these common trait-associated CNVs can vary substantially[15,16]. While most LEVs are rare, common risk variants with substantial effects are observed in Type 1 diabetes (i.e., the high-risk *HLA-DR* haplotype), breast cancer (i.e., *BRCA1* and *BRCA2* mutations)[17], and Alzheimer's Disease (i.e., *APOE4* haplotype).

The polygenic background in which a LEV is expressed may influence both its penetrance (i.e., the proportion of carriers with the disease phenotype) and expressivity (i.e., variation in phenotypic presentation). Certainly, under the liability-threshold model, liability derives from the cumulative effect of genetic and nongenetic risk and protective factors, and a critical threshold determines case status, which may indicate a testable pattern of relationships among these factors. Alternative models of disease liability may also be amenable to such statistical inference on the relationship between contributing factors[18]. It is reasonable to hypothesize that, for a range of complex disorders, those with the condition who derive little susceptibility to phenotype from their inherited common, small-effect variant polygenic burden (PB) may be more likely to harbor LEVs (i.e., sequence variants or CNVs, inherited or de novo). Conversely, the absence of LEVs in affected individuals would imply a greater role for the PB in conferring disease predisposition. We refer to this hypothesized inverse relationship between PB and LEV among cases as the "PB-LEV correlation" (Fig. 1). This hypothesized model is consistent with empirical data in several recent studies. For example, individuals with schizophrenia carrying a known LEV had a lower schizophrenia polygenic score (PGS) than those without such LEVs[19], although both groups had higher average schizophrenia PGS than controls. Similarly, Kuchenbaecker et al. found that breast cancer and ovarian cancer PGS are significantly associated with cancer risk even in *BRCA1* and *BRCA2* LEV carriers, suggesting a role for PGS in cancer risk management[20]. Similar observations for PGS were made for risk of breast and prostate cancer risk in male *BRCA1* and *BRCA2* mutation carriers[21]. Lee et al. reported that GWAS-identified loci modify the clinical onset of Huntington's disease, for which a rare CAG repeat on *HTT*[22] is causal. Finally, a recent study reported that the odds ratios (ORs) of LEV for coronary artery disease, breast cancer, and colon cancer are greater in high PGS quintile subjects than in low PGS quintile subjects[23]. However, the generalizability of these results to different disease risk models and different classes of genetic architectures remains unclear, and an analytic framework for further methodological and empirical investigations is lacking.

Here, we develop a summary-statistics-based framework to characterize and exploit the relationship between the PB and LEV among individuals sharing a diagnosis in clinically- and methodologically- relevant applications, with software implementation, PB-LEV-SCAN. This framework allows us to test several falsifiable hypotheses. Firstly, the framework raises the possibility that the common-variant PGS may provide a useful criterion for prioritizing samples for sequencing and rare variant discovery. Secondly, if implemented in clinical or research praxis[24], to what extent would different parameters—study-design, disease model, and genetic architecture—influence the result and be used as features for prediction? We apply our summary-statistics-based methodology in the context of empirical data, i.e., with parameter values observed in large-scale biobanks (see Methods), test the predictions and findings generated based on the models of the framework against empirical observations, and present a real-world application of the framework.

## Results

**Utility of PB-LEV correlation.** The PB-LEV correlation (Fig. 1) may provide a useful approach for probing the relationship between sources of genetic risk and for prioritizing samples for sequencing and rare variant discovery. Extensive simulations were therefore performed to calculate the utility (see Methods) of the PB-LEV correlation for each of the disease risk models and a genetic architecture chosen from the polygenic, negative selection, and LD-adjusted kinship models (Fig. 2). We use the term "utility", as the usefulness of the PB in some of the clinically important applications we have in mind (e.g., as a sample prioritization tool for sequencing or for well-powered discovery of a large-effect pathogenic variant) depends on our ability to detect the PB-LEV relationship from a (clinical) sample of causal variants and of individuals.

The models of disease risk included here are the liability-threshold model (in which a case is deterministically defined by exceeding a liability threshold) and the logit risk model (in which the definition of a case has a stochastic structure), each a joint model of how risk factors PB (denoted by $A$) and LEV (denoted by $R$) contribute to disease susceptibility. That is, if $p$ is the probability of disease risk, each model specifies $p$ as a function of the two sources of risk (see Methods):

$$p = g(A, R) \quad (1)$$

On the other hand, each genetic architecture models the distribution of the causal effect size $\beta_{PB,i}$ as a function of minor allele frequency $f_i$ and the extent of LD $l_i$ with neighboring variants as quantified by the LD Score (see Methods):

$$\beta_{PB,i} \sim \mathcal{N}(0, \psi(f_i, l_i)) \quad (2)$$

For example, the (baseline) polygenic architecture is defined by a fixed variance $\psi(f_i, l_i) = V$.

Specific simulation parameters such as the heritability of the trait attributable to the PB ($h^2_{PB}$), the heritability ($h^2_{LEV}$) explained by, and allele frequency ($f$), of the LEV, the disease prevalence ($K$), and sample size ($N$) (see Methods) were varied to assess the utility of the PB-LEV correlation.

We first considered the results under a liability-threshold model and the polygenic architecture as a baseline. Higher PB was, in general, observed for cases than for controls in the simulations (Supplementary Data 1). As expected, a larger sample size resulted in higher utility consistently across all simulation settings (Fig. 3). Higher trait variance explained by the LEV (equivalently $h^2_{LEV}$, when the outcome is standardized to $\mathcal{N}(0, 1)$) or higher common SNP-based heritability ($h^2_{PB}$) also resulted in higher utility (Fig. 3a, b), with the exception of the special case of very small sample size ($N = 500$). We then fixed $h^2_{LEV}$, $h^2_{PB}$, and disease prevalence, but varied the allele frequency of the LEV from 1e-4 to 0.05. The utility was negatively correlated with the allele frequency in the range of 0.001 to 0.05 (Fig. 3c). Because the

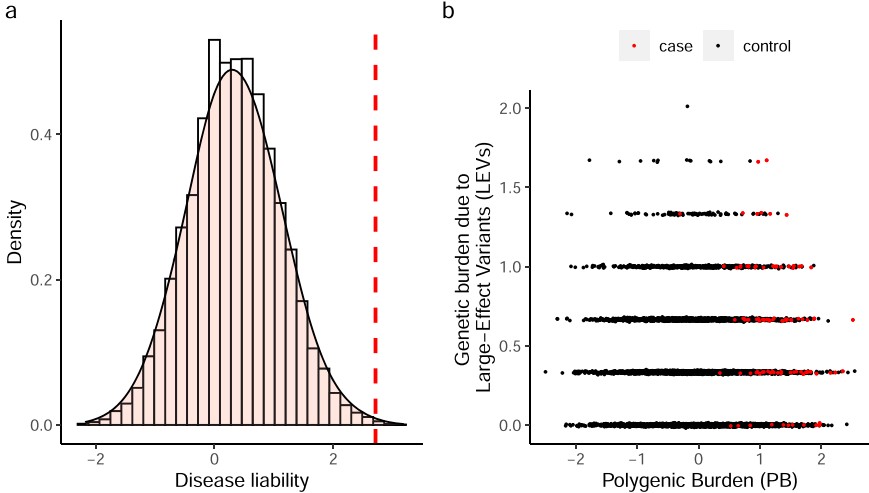

**Fig. 1 Central hypothesis underlying relationship between polygenic burden and large-effect (typically rare) variant.** For illustration here, we set the heritability due to the polygenic burden (PB), heritability due to large-effect variants (LEVs), and population prevalence to be 0.5, 0.1, and 0.01, respectively, and simulated $N = 10,000$ independent samples under the liability-threshold model of disease risk. Panel **a** shows the distribution of the underlying liability, a normally-distributed trait resulting from the cumulative effects of PB, LEVs, and the residual component. The red dashed line indicates the so-called liability threshold, which is determined by $K$, the population prevalence parameter. Panel **b** illustrates the relationship between the two orthogonal sources of genetic risk among individuals who share a diagnosis. Ideally, an inverse correlation of PB and LEV burden should be observed in cases (red dots, defined by the threshold line in panel (**a**)) under the liability-threshold model. In real-world data, the liability may include many additional effects not shown in this model. We refer to the inverse relationship between PB and LEV among cases (illustrated in panel (**b**)) as the "PB-LEV correlation". The PB-LEV correlation can then be used in downstream applications, e.g., predicting (for external validation) the proportion of LEV carriers among cases with a given PB profile, a key element of PB-based LEV screening. Source data are provided as a Source Data file.

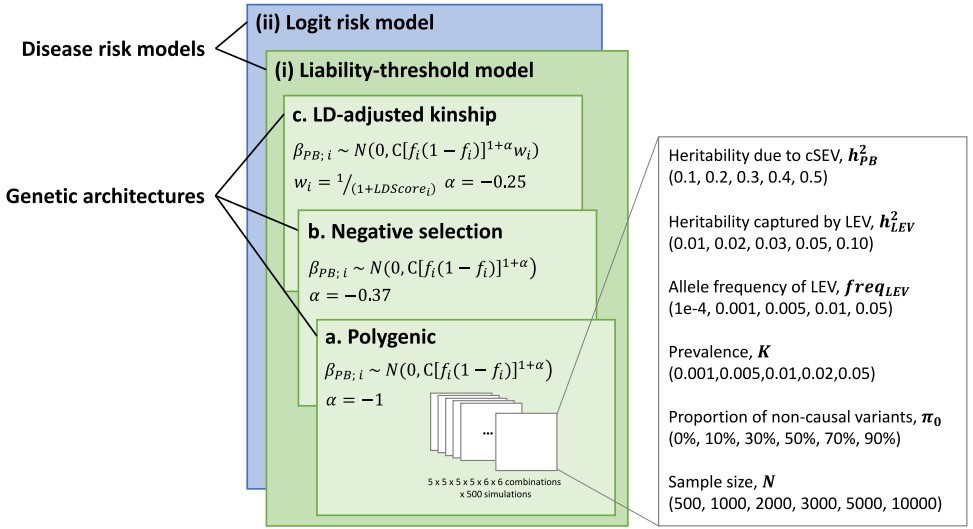

**Fig. 2 Generative models of disease liability and various genetic architecture models.** We calculated the disease risk probability assuming two models of disease liability, namely, the liability-threshold model (green panel) and logit risk model (blue panel). To simulate the effect sizes of common causal variants (and the resulting phenotypes), we considered three genetic architecture models: polygenic, negative selection, and LD-adjusted kinship. For each model of disease liability and model of genetic architecture, we varied the simulation parameters (the white panels), including the heritability $h^2_{PB}$ due to the common-variant polygenic burden (PB), heritability $h^2_{LEV}$ captured by the (primarily rare) LEV and its allele frequency $f$, disease prevalence $K$, and sample size $N$ (as a study-design parameter). The chosen values are shown in the brackets under each parameter. In simulated data, we calculated the utility of the PB-LEV correlation. In addition, varying the proportion of noncausal variants ($\pi_0$) in the estimate of PB, we quantified the power to detect the PB-LEV correlation. For each of the parameter combinations, we simulated 500 times to calculate the utility and power. These generative models of disease liability and genetic architectures provide the basis for a summary-statistics-based framework for inferences based on PB and LEV.

variance explained by the LEV was fixed, larger $f$ yielded smaller effect size $\beta_{LEV}$, implying that when the effect size difference between common and rare variants was smaller (i.e., the common and rare variants had more similar effects), the utility was lower. However, in the case of LEV allele frequency lower than 0.001, utility dropped, as the LEV might not be present in the sample.

For more common diseases (i.e., greater value of trait prevalence $K$), an increase in sample size resulted in an increase in utility (Fig. 3d).

We simulated alternative genetic architectures from the negative selection and LDAK models (see Methods). In contrast to the polygenic model, these genetic architectures assume that

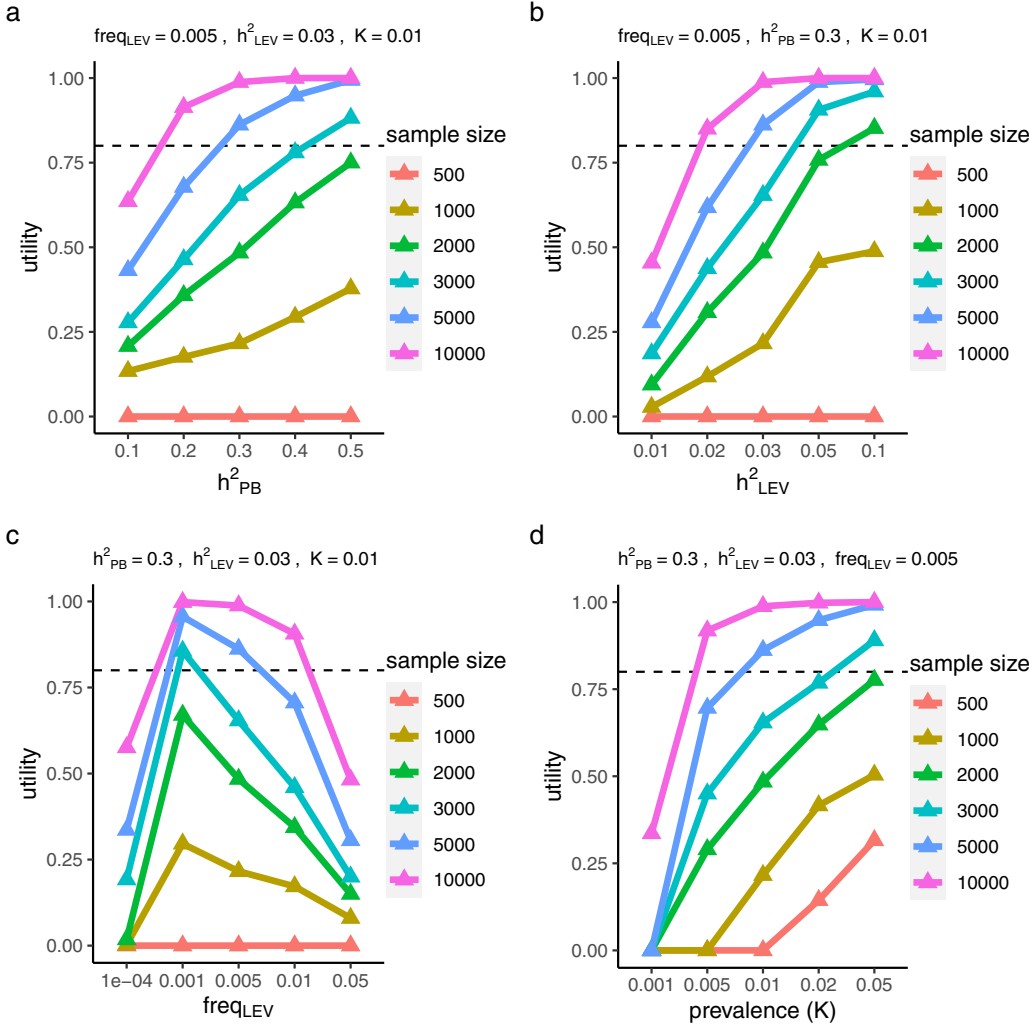

**Fig. 3 Utility of PB-LEV correlation under liability-threshold model.** We simulated the effect sizes for common and rare variants and calculated the probability of disease risk for each individual using real genotype data. Cases and controls were defined by the liability threshold. The utility is defined as the proportion of simulations for which, among cases, a significantly lower ($P < 0.05$) PB in LEV carriers than in noncarriers is observed. We varied the **a** heritability attributable to the common-variant polygenic component, **b** heritability captured by the LEV and **c** its allele frequency, and **d** disease prevalence while fixing all other parameters. Two-sample Wilcoxon test was performed to test whether PB was lower in LEV carriers than in noncarriers (one-sided test) in cases. The utility, to be contrasted with power, was calculated as the proportion of significant ($P < 0.05$) test results in 500 simulations, with different seeds for sampling causal variants and subjects. Broken line at 80% is a reasonable utility threshold. The results under the polygenic genetic architecture are shown. Results for the other two genetic architecture models can be found in Supplementary Fig. 1. Source data are provided as a Source Data file.

the contribution to $h^2_{PB}$ from the causal variants depends on the minor allele frequency or on both the minor allele frequency and the correlation with neighboring variants, respectively. The results on the utility of the PB-LEV correlation held robustly across these genetic architectures (Supplementary Fig. 1).

**Power**. In addition, under all three classes of genetic architectures, we simulated a mixture distribution consisting of null and causal effects (see Methods) to determine the degree to which a PGS with noncausal SNPs, as an estimate of the common-variant PB, might impact the detection of the PB-LEV correlation. The power significantly dropped when $\pi_0$ (the proportion of noncausal variants) increased while fixing a set of parameters ($h^2_{PB}$, $h^2_{LEV}$, $f$, and $K$) at "low" (Fig. 4a) and "high" levels (Fig. 4b). In 10,000 samples, the power (under "low" level) was substantially reduced from 98.2 to 94.6, 79.8, 48.0, and 14.8 when the proportion of noncausal variants was increased from 10 to 30, 50, 70, and 90, respectively. The power dropped below 50% for all the

cases when $\pi_0$ was set to 90%. The trend was consistent across the three genetic architectures (Supplementary Fig. 2).

Taken together, these results underscore the importance of determining, i.e., fine-mapping, the causal variants to be included in the PGS for the detection of the PB-LEV correlation. On the other hand, they also show that statistical power can be maintained at a sufficiently high level (i.e., at or above 80%) even when a large proportion of the variants (here, up to half) in the PGS are noncausal.

**Effect of negative selection on genetic architecture**. Negative selection has been proposed as a mechanism to explain the extreme polygenicity of complex traits. A genetic architecture constrained by negative selection would display "flattening" of the distribution of heritability[25], thereby showing more limited average per-allele effect for common variants across the genome. The action of negative selection could also induce effect size to vary with LD[26].

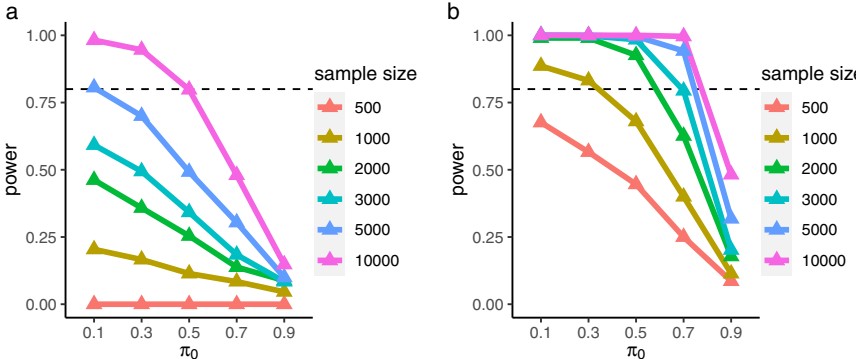

**Fig. 4 Power under liability-threshold model.** The proportion of noncausal common variants $\pi_0$ (0.1–0.9) in the common-variant polygenic component was considered in power estimation. For the other parameters, two different combinations were applied. We simulated **a** "low" ($h_{PB}^2 = 0.3, h_{LEV}^2 = 0.03, f = 0.005, K = 0.01$) and **b** "high" ($h_{PB}^2 = 0.5, h_{LEV}^2 = 0.1, f = 0.05, K = 0.05$) levels to show the effect of $\pi_0$ on the power at different levels. The results under the polygenic genetic architecture are shown. Results for the other two genetic architecture models can be found in Supplementary Fig. 2. Broken line at 80% is a reasonable threshold. Source data are provided as a Source Data file.

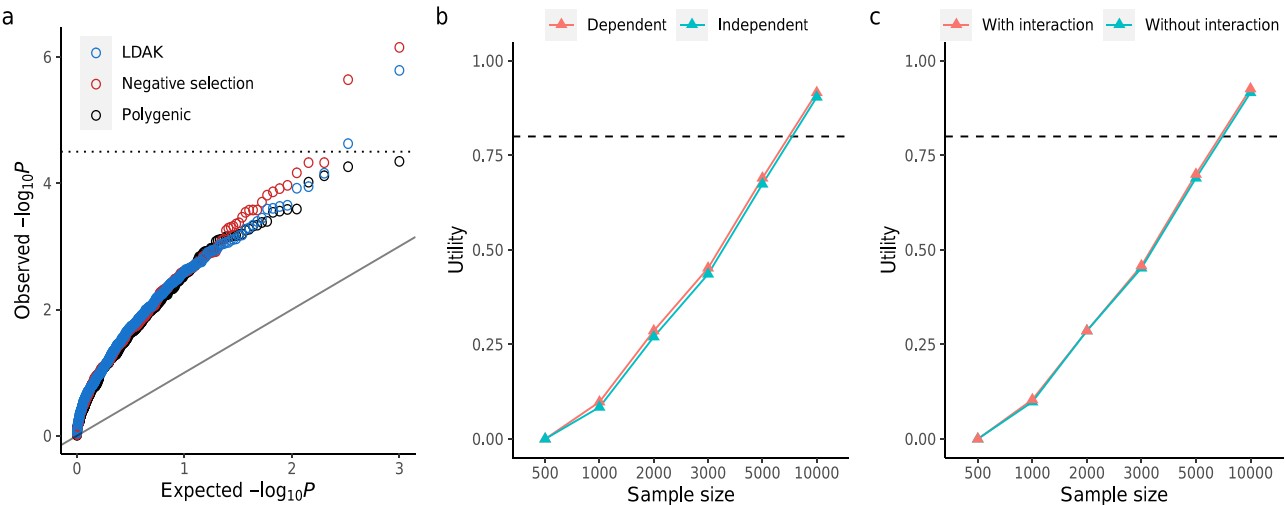

**Fig. 5 The PB-LEV correlation under different genetic architectures. a** We compared the significance of the correlation between the polygenic burden (PB) and large-effect variant (LEV) in cases for three classes of genetic architecture, namely, "polygenic", "negative selection", and "LDAK" (see Methods). The expected –log(P) and the observed –log(P) from 500 simulations are shown on x-axis and y-axis, respectively. In the case of relatively modest PB heritability ($h_{PB}^2 = 0.1$), higher-significance levels (i.e., lower p values) were observed for "negative selection" and the "LDAK" model (than for neutral "polygenic" architecture), both consistent with the "flattening" of heritability across the genome, which has been attributed to the action of negative selection. Here, we set $h_{LEV}^2, f$, and $K$ at $0.01, 0.001$, and $0.01$, respectively. Significance was assessed via Wilcoxon rank-sum test (one-sided). Given the specific combination of simulation parameter values, the dashed horizontal line at 4.5 is a potential threshold that can be used to rule out certain genetic architecture models through comparison of the observed PB-LEV correlation with the framework's predictions. **b** To test the effect of potentially dependent PB and LEV (in the general population) on the PB-LEV correlation (in cases), we assumed that the LEV was in high LD (D' = 1) with the top-ranked common variant (i.e., with the largest effect size) and compared the utility. Here, we set $h_{PB}^2 = 0.3, h_{LEV}^2 = 0.01, f = 0.001$, and prevalence (K) = 0.01, and assumed a polygenic genetic architecture and the liability-threshold model in the simulations. **c** To test the extent to which potential interactions between the PB and LEV could affect the PB-LEV correlation, we assumed that the interaction between the LEV and the common variant with the largest effect size contributed to the variance of the trait ($h_{interaction}^2 = 0.02$) and performed simulations. All other parameters were fixed as in (**b**). Broken line at 80% is a reasonable utility threshold. Source data are provided as a Source Data file.

We hypothesized that the influence of negative selection on genetic architecture could be tested through its impact on the PB-LEV correlation. Furthermore, this impact would be greater in the case of relatively modest PB heritability (with the same number of independently associated variants), which would indicate a more pronounced separation between average common-variant effects and the LEV. Consistent with this hypothesis, although all three classes of genetic architecture consistently showed a highly significant PB-LEV correlation throughout the simulations, the genetic architecture consistent with negative selection was more enriched for higher-significance PB-LEV correlation than the

(neutral) polygenic genetic architecture (Fig. 5a). Similarly, the LD-adjusted kinship model, which assumes an MAF-dependent genetic architecture (in addition to dependence on LD), had greater enrichment than the polygenic genetic architecture (Fig. 5a). Taken together, these results raise the possibility that certain types of genetic architecture can be ruled out for a given complex trait by the application of the framework.

**PB-LEV correlation in presence of disease subtypes.** We asked to what extent the presence of disease subtypes as defined by a heterogeneous PB effect alters our conclusions. The presence of

disease subtypes for a given complex disease is typically a priori not known. By varying the heterogeneity of effects across subtypes and the proportion of minor subtype cases (see Methods), we observed highly consistent results (Supplementary Fig. 3). This analysis shows the robustness of the relationship between LEV and PB in cases even in the presence of disease subtypes.

**Robustness to independence and additivity of PB and LEV.** All simulations thus far assumed that the PB and LEV are independent (in the general population) and that their contributions to risk are additive. Here, we simulated two additional scenarios wherein (1) small-effect common variants and LEV are dependent and (2) the interaction between small-effect common variants and LEV also plays a role in disease risk.

For the first scenario of dependence, we assumed that the LEV was in high LD ($D' = 1$) with the common variant with the largest effect size. We performed simulations and compared the utility of the PB-LEV correlation with the utility when PB and LEV were independent. For the simulations, we set $h_{PB}^2 = 0.3$, $h_{LEV}^2 = 0.01$, $f = 0.001$, and prevalence ($K$) = 0.01, and assumed a polygenic genetic architecture and the liability-threshold model. The values for the utility were similar for the dependent and independent cases as sample size was varied from 500 to 10,000 (Fig. 5b).

For the second scenario of interaction, we assumed that the interaction between the LEV and the common variant with the largest effect size contributed to the trait variance ($h_{interaction}^2 = 0.02$). For all other parameters, we used the same settings as above. Here again, the utility values between the disease models with and without interaction were similar (Fig. 5c).

**Application of PB to identify at-risk individuals comparable to LEVs.** For clinical application, it is of considerable interest to determine, from simulations, to what extent one can use the common-variant PB to identify at-risk individuals with PB comparable to large-effect mutations. We calculated the OR of the LEV (by taking LEV carriers and noncarriers as the exposed and the unexposed group, respectively) and the OR of PB (by taking the top 1, 5, and 10% of the PB-ranked samples as the exposed group and the remaining samples as the unexposed group). For our assumed simulation settings (under a polygenic genetic architecture), the OR of the LEV and OR of the PB showed a similar order of magnitude. The OR of the LEV and that of the PB tended to converge as $h_{PB}^2$ increased (Fig. 6a) or as the heritability due to the LEV decreased (Fig. 6b). Assuming $h_{PB}^2$ to be 0.3, we observed that the OR of the LEV was comparable to the OR of the PB (using any of the three PB cutoffs) when $h_{LEV}^2$ was in the range between 0.01 and 0.03 (Fig. 6b). However, as $h_{LEV}^2$ increased, the two ORs would increasingly diverge. Nevertheless, at least based on the estimates of $h_{LEV}^2$ from available empirical data in a large-scale biobank (UK Biobank), the condition $h_{LEV}^2 > 0.03$ would appear to be rather uncommon (and indeed the maximum $h_{LEV}^2$ estimate was 0.025 among the 36 heritable traits with an identified LEV, from studies using the UK Biobank[27,28] [Supplementary Data 2 and Methods]). Thus, our simulations identified a specific range of these parameters wherein the OR of the LEV and that of the PB would be expected to be similar. As expected, lower LEV allele frequency would result in increased OR of the LEV (Fig. 6c). The OR of the LEV and that of PB tended to become more similar with higher disease prevalence, i.e., for more common diseases (Fig. 6d). Similar patterns were observed for the other two classes of genetic architectures (Supplementary Figs. 4, 5).

Our analyses so far assumed that all causal variants were known (i.e., $\pi_0 = 0$) for PB estimation. To fit realistic situations,

we varied the proportion of noncausal variants ($\pi_0$) from 0 to 0.9 and reestimated the OR. The OR estimate both in carriers and in noncarriers substantially decreased when the $\pi_0$ was increased (Supplementary Fig. 6). The difference in OR for PB between carriers and noncarriers was no longer detected when 90% of the variants used for the PB were noncausal. Similar results were observed under the other two classes of genetic architectures.

We compared the change in OR of the PB (per sd change) between LEV carriers and noncarriers while varying each of $h_{PB}^2$, $h_{LEV}^2$, $f$, and $K$, i.e., we compared the slopes $\frac{\partial(OR)}{\partial h_{PB}^2}$, $\frac{\partial(OR)}{\partial h_{LEV}^2}$, $\frac{\partial(OR)}{\partial f}$, and $\frac{\partial(OR)}{\partial K}$, respectively, between LEV carriers and noncarriers. The 2.5–97.5 percentile range for the OR among LEV carriers and noncarriers overlapped for each simulation setting (Fig. 6e–h, under polygenic genetic architecture). No difference in OR was observed between LEV carriers and noncarriers (i.e., no evidence of interaction between PB and LEV), which is consistent with the simulation's assumed additivity of effects but also with recent empirical studies for some traits[20,23,29]. However, for a given change in $h_{PB}^2$, our framework would predict that change in OR of the PB should be generally higher in noncarriers than carriers (Fig. 6e) while a more limited differential change in OR between carriers and noncarriers was observed by varying $h_{LEV}^2$, $f$, and $K$ (Fig. 6f–h). Thus, among the tested parameters, $h_{PB}^2$ is the most important determinant of how differently, between carriers and noncarriers, the OR of the PB changes. We observed similar results for the other two classes of genetic architectures (Supplementary Figs. 7, 8).

**Simulations using alternative model of disease risk.** We considered an alternative model of disease liability, namely the logit risk model. Under this model, a case is defined by a stochastic draw (in contrast to the deterministic cutoff of the liability-threshold model) from binomial distribution (using the probability of disease risk). In general, the results from the logit risk model were consistent with those from the liability-threshold model (Supplementary Figs. 9–13).

**Testing the PB-LEV correlation in real data.** We found that Tourette Syndrome (TS) patients[30,31] harboring LEVs (i.e., de novo LoF/Mis3 coding variants or lrgCNVs, Supplementary Table 1 and Supplementary Table 2; $n = 36$) had significantly lower common-variant PGS than TS patients ($n = 480$) without such chromosomal events ($P = 0.020$, Supplementary Table 3). The PGS$_{TS}$ accounted for 3.6% of the total variance in lrgCNV carrier status according to Nagelkerke's $R^2$ calculation. In contrast, among cases with Obsessive-compulsive disorder (OCD)[30] ($n = 919$), no significant difference ($P = 0.120$) of PGS$_{OCD}$ was observed between LEV (lrgCNVs) carriers ($n = 61$) and noncarriers ($n = 858$).

We excluded chromosome 6 to avoid any possible confounding by complex LD patterns in the HLA region for PGS of type 1 diabetes T1D. Individuals ($n = 873$) who were carriers of the high-risk alleles at the *HLA-DRB1* locus[32] (high-risk genotypes = 3/3, 3/4, or 4/4) had significantly lower covariate-adjusted PGS$_{T1D}$ than individuals ($n = 985$) who were carriers of the lower risk alleles (low-risk genotypes = 3/not [3 or 4], 4/not [3 or 4], or not [3 or 4]/not [3 or 4], Wilcoxon rank-sum test $P = 0.007$).

**Application to large-scale biobank.** Using empirical data with various levels of heritability, disease prevalence, and LEV frequency, as observed in the UK Biobank, we sought to (1) evaluate whether the predictions generated based on the models of the framework can be confirmed in empirical

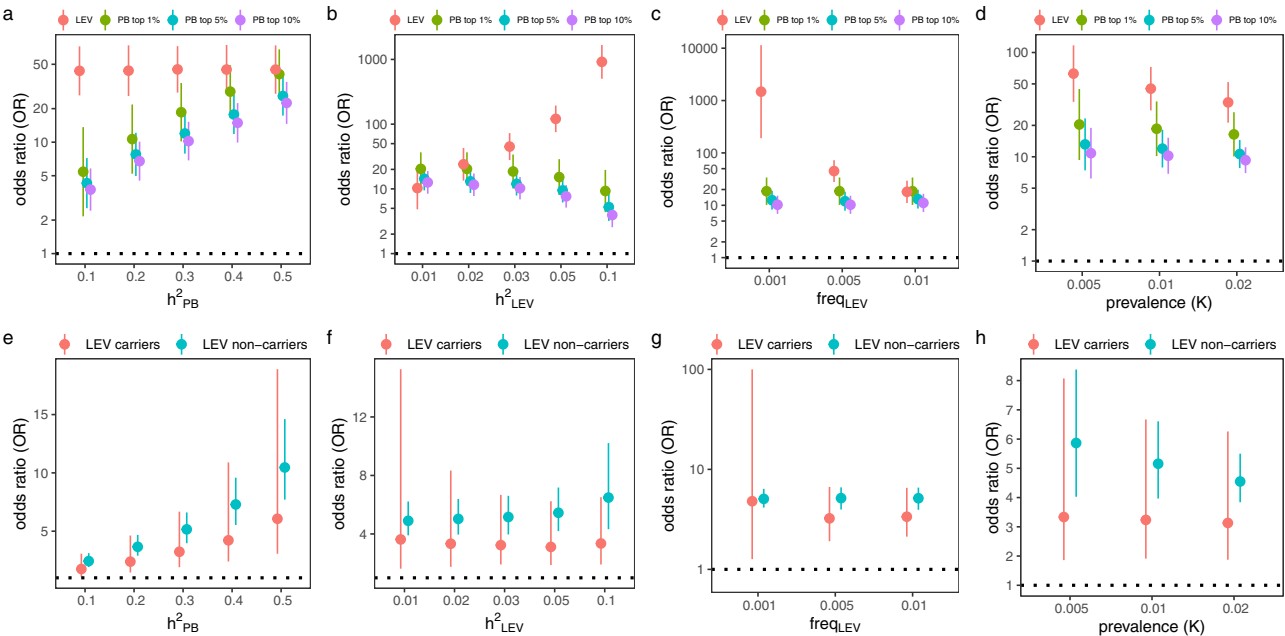

**Fig. 6 The odds ratio (OR) of small-effect, common variant-based polygenic burden. a–d** OR comparison between the large-effect variant (LEV) and polygenic burden (PB). One potential application of our framework is to identify at-risk individuals with PB similar to a LEV. Thus, we investigated the scenarios in which the OR for the LEV and for the PB are comparable. We calculated the OR of the LEV under the liability-threshold model and the polygenic genetic architecture while varying the common SNP-based heritability ($h_{PB}^2$), the heritability of LEV ($h_{LEV}^2$), the allele frequency of LEV ($f$), and the prevalence ($K$) and assuming 10,000 independent samples. The OR of the PB was determined by comparing individuals with high-PB (top 1, 5, and 10% of distribution) with the remainder of the population. The point estimates and the 95% confidence interval (CI) are shown as dots and horizontal lines, respectively. **e–h** Change in the OR of PB (per sd change) with respect to change in parameter differs between LEV carriers and noncarriers. The OR of the PB was calculated under the liability-threshold model and the polygenic genetic architecture while varying $h_{PB}^2$, $h_{LEV}^2$, $f$, and $K$. In simulations, we assumed 10,000 samples. The median of the OR across simulations is shown as a dot, while the 2.5th and 97.5th percentile of the OR across simulations are represented by the horizontal segments. Among the parameters tested here, the $h_{PB}^2$ is the most important determinant of how differently, between carriers and noncarriers, the OR of the PB changes, as can be seen from the "slope" at each point. In panels (**a**) and (**e**), we fixed $h_{LEV}^2$, $f$, and $K$ at 0.03, 0.005, and 0.01, respectively, while varying $h_{PB}^2$. In panels (**b**) and (**f**), we fixed $h_{PB}^2$, $f$, and $K$ at 0.3, 0.005, and 0.01, respectively, while varying $h_{LEV}^2$. In panels (**c**) and (**g**), we fixed $h_{PB}^2$, $h_{LEV}^2$, and $K$ at 0.3, 0.03, and 0.01, respectively, while varying $f$. In panels (**d**) and (**h**), we fixed $h_{PB}^2$, $h_{LEV}^2$, and $f$ at 0.3, 0.03, and 0.005, respectively, while varying $K$. The horizontal broken line at OR = 1 shows the null. Source data are provided as a Source Data file.

observations and (2) illustrate real-world applications of the framework. In total, 36 heritable traits, each with at least one identified LEV, were included (Supplementary Fig. 14). Most of the UK Biobank traits reflected the simulation parameters (Supplementary Data 2). Twenty-five percent (9 of 36) of the traits showed a high utility (≥80%) of the PB-LEV correlation (Supplementary Data 2).

We considered the probability of LEV carriers among cases, $P(\text{LEV} \mid Y = 1)$, estimated here from simulations as the number of LEV carriers per 1000 cases, to quantify the effectiveness of the framework for prioritizing cases for sequencing studies. We estimated the number of LEV carriers per 1000 cases for each of ten equally-sized bins of PB risk score. We found that cases from the lowest PB risk bins tended to have the highest proportion of LEV carriers. As an example (Fig. 7a), we would expect to find 27 cases carrying a stop-gain mutation (OR = 5.1, MAF = 0.0013) on *MYOC* per 1000 glaucoma cases from the lowest PB risk bin. However, among the 1000 glaucoma cases in the highest PB risk bin, we would expect to find only six carriers. We analyzed four additional traits (Fig. 7b–e) representing three distinct patterns. The results on malignant melanoma (Fig. 7b) indicated that only cases with low-PB risk would be worth sequencing for LEV screening. For both Crohn's disease (Fig. 7c) and acute tonsillitis (Fig. 7d), the striking difference in the estimated proportion of LEV carriers among cases from the different PB risk bins would suggest the usefulness of PB-based prioritization. Per 1000

Crohn's disease patients, 214 cases would be expected to be *NOD2* frameshift mutation carriers from the lowest PB risk group whereas only about 67 cases from the highest PB risk group would be expected to be carriers of the mutation (Fig. 7c). Nearly half of the acute tonsillitis cases (452/1000) from the lowest PB risk group were predicted to be *OXCT2* missense mutation carriers, which was a nearly threefold increase relative to the highest PB risk group (Fig. 7d). However, PB-based prioritization appears to be less helpful for screening low frequency, large-effect type 2 diabetes-associated variants (Fig. 7e).

We further compared the predictions from our framework with observations from the empirical data: breast cancer, colorectal cancer, type 2 diabetes, and short stature[28]. Under the liability-threshold model applied to these traits, our framework would predict that low-PB cases tended to have a higher probability of carrying an LEV than high-PB cases (the median of PB was used to define low and high-PB groups). The framework's predicted distribution of the proportion of LEV carriers was generated (shown as a boxplot in Fig. 8a). Notably, the empirical data (i.e., the actual proportion of LEV carriers, marked as a red diamond) were concordant with the predictions. In addition, we induced different levels of noise for the PB by varying the proportion of noncausal variants in the polygenic risk score, $\pi_0$ (evaluated at 0, 0.5, and 0.8). The pattern of low-PB cases having a higher proportion of LEV carriers still held despite the presence of induced noise although the difference between low-PB cases

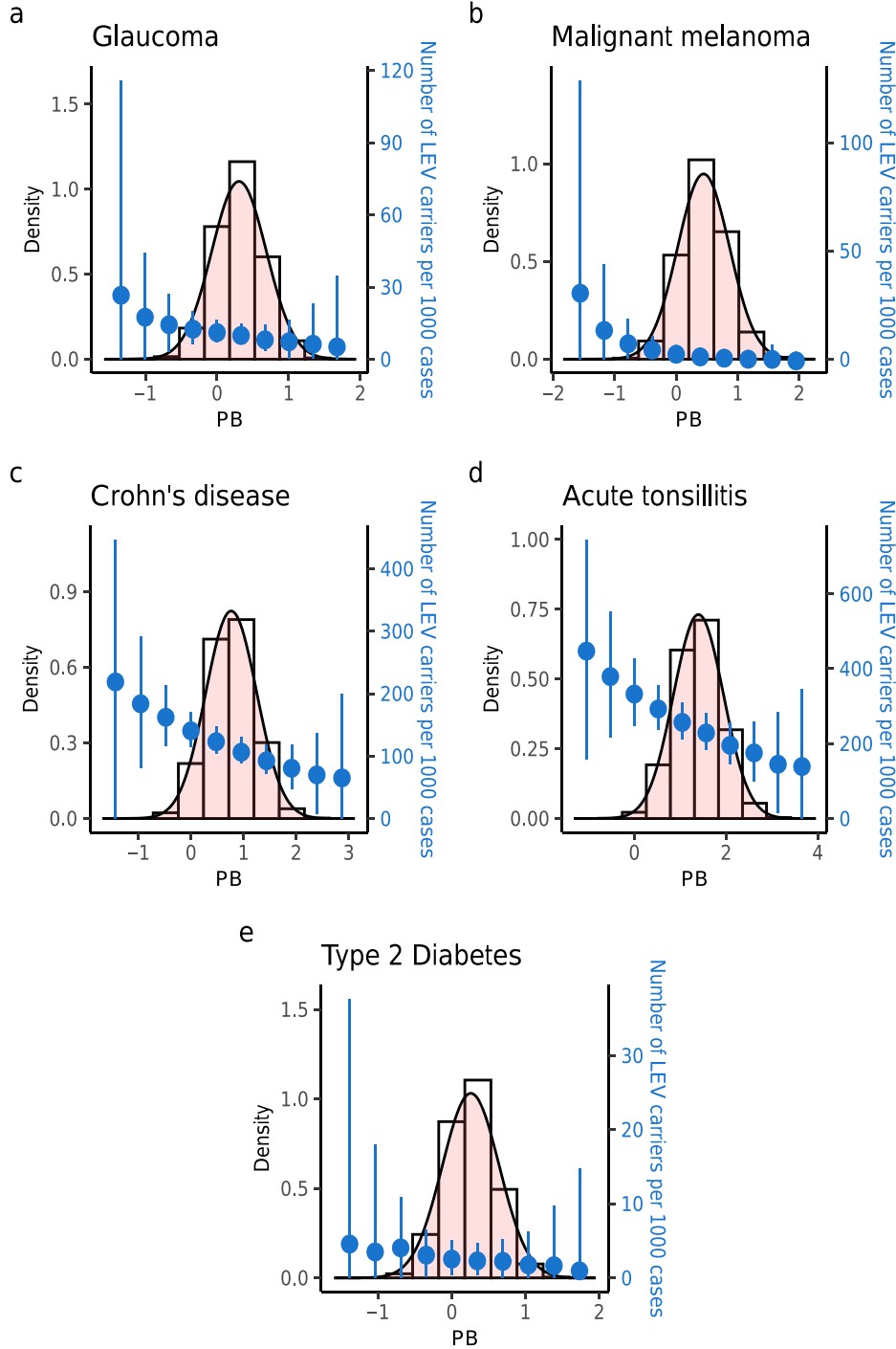

**Fig. 7 Cases with low polygenic risk score have higher probability of carrying an LEV.** Based on empirically-informed parameter values (i.e., derived from the UK Biobank), we performed simulations (under the liability-threshold model and the genetic architecture in line with negative selection) and compared the number of LEV carriers per 1000 cases among the different polygenic risk scores for **a** glaucoma, **b** malignant melanoma, **c** Crohn's disease, **d** acute tonsillitis, and **e** type 2 diabetes. For each trait, we grouped the cases into ten equally-sized polygenic risk score bins. For each bin, the mean ± 1sd of the number of LEV carriers per 1000 cases is displayed as a blue circle and bar. The distribution of polygenic risk score is shown as a histogram. For example, per 1000 Crohn's disease cases, 214 would be expected to be *NOD2* frameshift mutation carriers from the lowest PB risk group; in contrast, PB-based LEV screening would be less effective for type 2 diabetes. The proportion of LEV carriers in the lowest PB risk group for acute tonsillitis was a nearly threefold increase relative to the highest PB risk group. Thus, the framework (assuming prespecified simulation parameters) provides testable predictions on the number of LEV carriers per 1000 cases with a given PB profile and on the sample polygenic risk score profile to optimize LEV screening. Detailed results can be found in Supplementary Data 2.

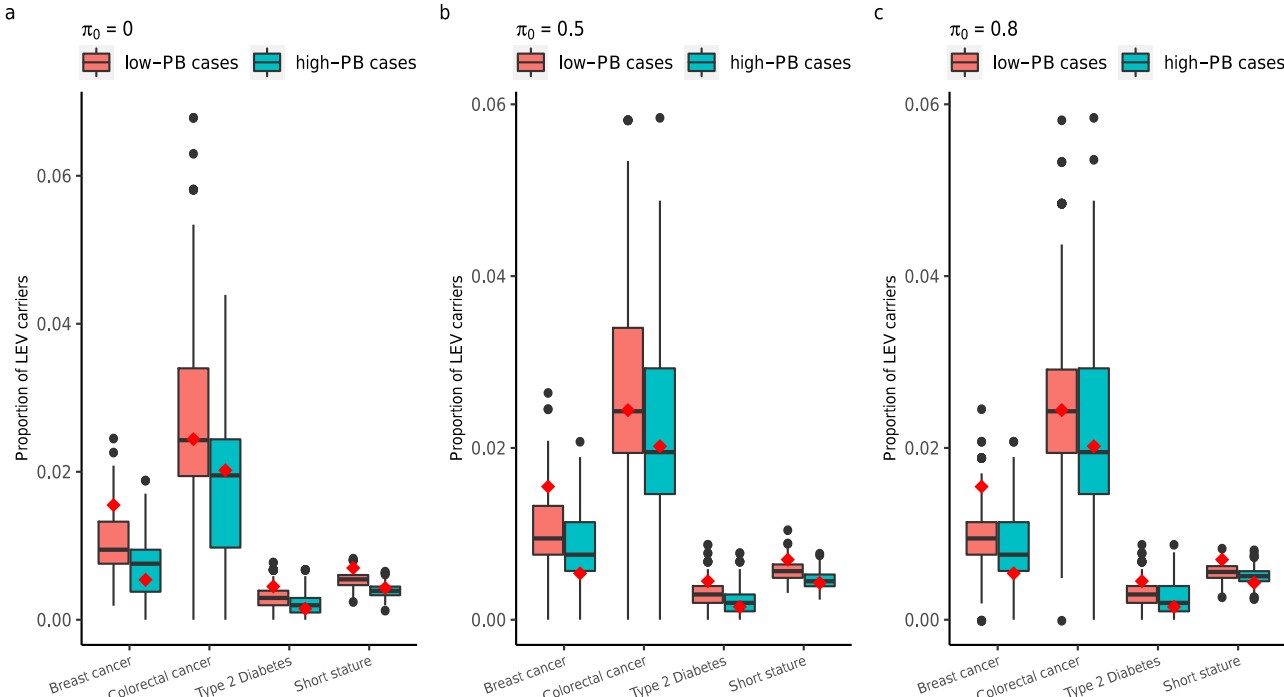

**Fig. 8 Prediction of the framework on proportion of LEV carriers in cases with a given PB profile matches empirical observations, providing an LEV screening approach.** Using empirically-informed values (i.e., derived from the UK Biobank) of the parameters (including the common variant-based heritability; disease prevalence; allele frequency and odds ratio of the LEV), we performed simulations for four traits under the genetic architecture in line with negative selection. We varied the $\pi_0$ (the proportion of noncausal variants in the polygenic risk score) at **a** 0, **b** 0.5, and **c** 0.8. Cases were defined under the liability-threshold model and classified into "low-PB" and "high-PB" (using the median as the cutoff) groups. The proportion of LEV carriers in each group was estimated as the number of cases carrying the LEV divided by the total number of cases. The distribution of the proportion (across 500 simulated sets) is shown in the boxplot. The median of the proportion is visualized as a black segment in the middle of the box. The lower and upper hinges correspond to the first and third quartiles (the 25th and 75th percentiles). The upper/lower whisker extends from the hinge to the largest/smallest value no further than/at most 1.5 * IQR from the hinge (where IQR is the inter-quartile range or the distance between the first and third quartiles). The actual observed proportion of LEV carriers for each PB profile from empirical data (with matching parameters as the simulations) is marked as a red diamond. Thus, the prediction of the framework and the empirical dataset (with matching parameters as the simulations) were concordant. Source data are provided as a Source Data file.

and high-PB cases became smaller with higher induced noise (Fig. 8b, c). Notably, the concordance between predicted and observed proportion remained. Under the logit risk model, we also observed consistent results between the framework's predictions and the empirical data for breast cancer, type 2 diabetes, and short stature (Supplementary Fig. 15). For colorectal cancer, a higher proportion of LEV carriers was predicted in low-PB cases, in agreement with empirical data; however, the predicted proportion of LEV carriers in both low-PB and high-PB groups was slightly higher than observed from the empirical data (Supplementary Fig. 15).

Having observed striking concordance between the predictions and empirical data, we applied our summary-statistics-based framework to all remaining UK Biobank traits to generate additional predictions. The results consistently showed the value of PB-based LEV screening for almost all of the tested traits, covering a diverse range of levels of PB heritability, disease prevalence, and LEV allele frequency (Supplementary Fig. 16). On average, we found the proportion of LEV carriers among cases with the lowest PB risk (i.e., the first PB risk bin) to be 2.6 times (range: 1.4–32.6) as large as the proportion among cases with the highest PB risk (i.e., the tenth PB risk bin). Detailed results can be found in Supplementary Data 2.

## Discussion

This study provides a framework for integrative analysis of large-effect genetic risk factors and their polygenic background. We

evaluate the impact of several parameters (i.e., heritability, disease prevalence, LEV frequency) under different models of disease liability and diverse classes of genetic architectures on the utility and power of the PB-LEV correlation. The framework makes certain predictions (e.g., reduced PGS stratification for LEV carriers compared with noncarriers) and presents several testable, falsifiable hypotheses (e.g., estimating the proportion of LEV carriers among cases with a particular PB profile given study-design, disease model, and genetic architecture parameters informed by empirical genomics data). Using empirical data, we confirmed the signature of the PB-LEV correlation in a large-scale biobank and, moreover, observed a significant PB-LEV correlation in early-onset phenotypes such as TS and T1D. In addition, we show that, for a set of choices of the parameters, the proportion of LEV carriers among cases given a PB profile can be accurately predicted, with clear utility for clinical and sequencing applications.

As a baseline, we assumed a liability-threshold model and the polygenic architecture. Consistent results were observed when we varied the disease liability and genetic architecture models. We found that the action of negative selection on the genetic architecture[25] might be detected through its more pronounced effect on the PB-LEV correlation in comparison with the neutral polygenic genetic architecture.

To explore the clinical relevance of our framework, we compared the OR of the PB and that of the LEV, uncovering a range of conditions in which the risk derived from the PB is comparable

to the LEV-derived risk. A recent study reported confirmatory results in empirical data (including CAD, Atrial fibrillation, T2D, IBD, and breast cancer GWAS data)[33].

Although some recent studies have provided support for the notion that, for many complex traits, PB and LEV are independent and additive[20,23,29], we examined two additional scenarios in which the assumption of either independence or additivity is relaxed. For the scenario of dependence, since we assumed that all causal common variants were independent, assuming $D' = 1$ between LEV and the common variant with the largest effect size is an extreme case. For the potential interaction of PB and LEV, we assumed a non-negligible role of interaction (i.e., explaining twice as much of the trait variance as the marginal LEV effect) in disease risk. In both scenarios, simulation results showed that the PB-LEV correlation held.

It is possible that the nonsignificant results for OCD may be due to implicit or chance ascertainment schemes in which cases were enriched for polygenic etiology (i.e., probands taken from multiplex families), consequently decreasing the power to identify pathogenic rare variants within the sample and negatively impacting the power to detect the PB-LEV correlation. Notably, we did not explicitly model gene networks or pathways shared between PB and LEV, i.e., biological correlation, which may explain the result for OCD. Certainly, replication in larger samples with a more comprehensive model (e.g., as implemented in CORE GREML[34]) will be needed to conclusively determine the strength of the correlation in all complex traits tested to date.

As expected, our analysis revealed significantly lower polygenic risk score in T1D cases with high-risk HLA-DRB1 genotypes. This analysis provides another confirmation that known variants with large effect—even when they are not fully penetrant—may exhibit a negative correlation with common-variant polygenic risk within a patient population.

Given only summary-level data for key parameters (e.g., trait heritability and LEV effect size), our framework provides an accurate estimate of the proportion of LEV carriers among cases with different PB profiles. Leveraging empirical data from the UK Biobank, we quantified the performance (i.e., the accuracy of the estimate) by comparing the predicted and observed results. Notably, the effectiveness of utilizing the PB risk for LEV screening varied by trait. For malignant neoplasms (malignant melanoma, basal cell carcinoma, and colorectal cancer) and diverse trait classes, including inflammation (Crohn's disease, regional enteritis, acute tonsillitis, phlebitis, and thrombophlebitis), cognition (delirium), circulation (acute myocardial infarction), and metabolism (gout), our results show that it is more effective to identify LEV carriers among cases with low-PB. In contrast, for type 2 diabetes and cellulitis, the effectiveness of using the PB risk is reduced. Notably, the framework's predictions were confirmed by empirical data in an external study[28]. It should be noted that the current framework's predicted proportion of LEV carriers among cases with a given PB profile may underestimate the true value. Our empirical understanding of LEVs is limited: (1) noncoding LEVs are not covered in exomesequencing data and (2) for each trait, only the top-ranked LEV was considered in this study.

Recent studies have shown that proxy measures of PB, such as family history of neuropsychiatric phenotypes, are also significantly negatively associated with LOF de novo variant rate in ASD samples providing additional support for the hypothesis that in the absence of common-variant polygenic risk, rare pathogenic de novo, or inherited variants must be more salient risk factors[35]. Our study suggests that PB is a useful addition to the prioritization schemes of samples likely to harbor rare pathogenic variants, particularly in the absence of extensive family history information. Since rare pathogenic variants may be enriched for

loss-of-function and since the genetic dose-response curve may be extrapolated from the regulatory range to loss-of-function[36], genetically determined expression[37,38] may be used to further investigate the PB-LEV correlation. Furthermore, cases with low common-variant polygenic risk and no apparent rare variant burden who are nonetheless affected, may represent a subgroup of cases enriched for epigenetic or environmental risk factors. This raises the possibility of a "stratified medicine" strategy based on PB. It follows that genetic epidemiology studies that include common-variant polygenic and rare variant risk scores in their models may further increase their power to identify shared and modifiable environmental risk factors.

The current study has several limitations. The disease models we used are plausible and wide-ranging, but they are not likely to cover all possible scenarios. Although we tested relaxing the assumptions of additivity and interaction of PB and LEV, comprehensive studies of generative models are needed. Diseases are complex, involving heterogeneous causal etiologies for individual variants and implicated biological processes and pathways, which the framework may not accurately model. Although we validated the predictions of the framework in several disease datasets in a large-scale biobank, the generalizability of the framework needs to be confirmed in additional datasets.

Several future applications come to mind. First, as we noted in Methods, estimation of disease prevalence using the genetic risk score follows from the framework. For highly heterogeneous traits, the less noisy genetic risk score may lead to improved estimation. Second, the framework raises the possibility of obtaining an estimate of the proportion of noncausal variants ($\pi_0$) in the PB. We provide a software implementation, PB-LEV-SCAN, to enable application of the framework to other complex traits and the testing of other hypotheses in future studies.

In conclusion, we developed a summary-statistics-based framework that utilizes the relationship between PB and LEV among cases for some methodologically- and clinically-relevant applications, including PB-based LEV screening. The framework's testable, falsifiable predictions on the proportion of LEV carriers among cases with a given PB profile under prespecified simulation parameters were confirmed by empirical data. Application of the framework to a large-scale biobank showed that the effectiveness of PB-based LEV screening varied substantially by trait. Taken together, these findings shed critical light on the use of PB in clinical practice.

## Methods

**Assumptions of the framework**. The first major assumption of the mathematical framework is the choice of generative model of disease liability. The second major assumption concerns the genetic architecture. Here, initially, we assume that the effects of the (genome-wide) PB and the LEV are independent and additive in defining the disease liability. For sensitivity analysis, we also considered the scenario in which the assumption of independence or additivity is relaxed. Let $h_{PB}^2$ and $h_{LEV}^2$ be the heritability due to the common variant-based polygenic background and the LEV, respectively.

**Generative model of disease liability**. Under the liability-threshold model (Fig. 1), individuals whose liability ($L$) exceeds a given threshold ($t$) are cases. If the effects of PB and the LEV are independent and additive, the liability can be written as:

$$L = A + R + e \tag{3}$$

where $A$ is the additive effects of PB:

$$A \sim \mathcal{N}(0, h_{PB}^2) = \sqrt{h_{PB}^2} * \mathcal{N}(0, 1) \tag{4}$$

$R$ is the large-effect variant (LEV) burden with effect size $\beta_{LEV}$ and allele frequency $f$:

$$R \sim (\beta_{LEV})^2 * \text{Binomial}(2, f) \tag{5}$$

and $e$ is the residual component:

$$e \sim \mathcal{N}(0, 1 - h_{\text{PB}}^2 - h_{\text{LEV}}^2) \qquad (6)$$

Here we assume a single LEV (see, however, below in case of LEV heterogeneity). The disease risk probability $p$ thus satisfies:

$$p = P(L > t) = P(e > t - A - R) = \Phi\left(-\frac{(t-x)}{\sqrt{1 - h_{\text{PB}}^2 - h_{\text{LEV}}^2}}\right) := \Phi^*(x) \qquad (7)$$

where $\Phi$ is the cumulative distribution function (CDF) of the standard normal $\mathcal{N}(0, 1)$ and $x = A + R$ is the total genetic risk. Note that the liability $L = x + e$ is unimodal. The threshold $t = \Phi^{-1}(1 - K)$ is a function of disease prevalence $K$, by definition. Note $\Phi^*$ is a composition of the linear function $m(x) = \frac{x - \Phi^{-1}(1-K)}{\sqrt{1 - h_{\text{PB}}^2 - h_{\text{LEV}}^2}}$ and the CDF $\Phi$. This implies that $\Phi^*$ is an increasing function of the genetic risk score $x$:

$$\frac{\partial p}{\partial x} = \frac{\partial \Phi}{\partial m}\frac{\partial m}{\partial x} = \phi\left(\frac{x - \Phi^{-1}(1-K)}{\sqrt{1 - h_{\text{PB}}^2 - h_{\text{LEV}}^2}}\right)\frac{1}{\sqrt{1 - h_{\text{PB}}^2 - h_{\text{LEV}}^2}} \qquad (8)$$

where $\phi$ is the probability density function of the standard normal distribution.

An alternative model is a logit risk model. Here the disease risk probability is connected to the total genetic risk $x = A + R$ via the function:

$$p = 1/(1 + e^{-(u_1 x + u_0)}) := \text{logit}^{-1}(u_1 x + u_0) \qquad (9)$$

Therefore, the total genetic risk can be written in terms of the log odds ratio: $u_1 x + u_0 = \ln\left(\frac{p}{1-p}\right)$, where the intercept term $u_0$ is a population-level parameter. While we use the same notation $x$ for genetic risk in the liability-threshold model and in the logit risk model, the genetic risk is modeled as a random effect for the former and a fixed effect for the latter. However, for the analysis of interest to us (which is not variance estimation), this distinction does not play a role, and thus we use the same notation. We note that the genetic risk score $x_{\text{logit}}$ under the logit risk model maps to the same disease risk probability (i.e., $p = p_{\text{logit}} = p_{\text{liability-threshold}}$) as the genetic risk score $x_{\text{liability-threshold}}$ under the liability-threshold model given by the following:

$$x_{\text{liability-threshold}} = (\Phi^*)^{-1}\left(\text{logit}^{-1}\left(u_1 x_{\text{logit}} + u_0\right)\right) \qquad (10)$$

Under the logit risk model, the case status of an individual has a stochastic structure:

$$Y \sim \text{Binomial}(1, p) \qquad (11)$$

We note that we can derive the prevalence $K$ as follows:

$$K = P(Y = 1) = \int_0^1 p\varphi(p)dp = \mathbb{E}[\mathbb{p}] \qquad (12)$$

where $\varphi(p)$ is the probability density of $p$ and $\mathbb{E}$ is the expectation operator. As the disease risk probability $p$ is a function of the genetic risk score $x = A + R$, the framework enables estimation of disease prevalence $K$ using the genetic data (which may be less noisy than the phenotype information).

In addition to the assumption of independent, additive effects of PB and LEV, as captured in the representation of the total risk score $(x = A + R)$, we investigated other formulations of the total risk score, wherein either independence or additivity of effects is violated. To relax the assumption of independence, we considered the case wherein the LEV was in high LD $(D' = 1)$ with the common variant with the largest effect size. To relax additivity, we assumed that the interaction between the LEV and the common variant with the largest effect size contributed to the trait variance. In this case, the liability has a nonzero interaction effect:

$$L = A' + R + A^*R + e \qquad (13)$$

where $A^*$ and $A'$ denote the effect of the common variant with the largest effect size and the effect of the remaining common variants, respectively. The $A^*R$ denotes the interaction effect.

**Disease subtypes.** We investigated the PB-LEV correlation in the presence of disease subtypes. Here, we assumed two subtypes, hereafter called major and minor, without loss of generality. We assumed a linear model[39] of the underlying liability that reflects the presence of subtypes using a polygenic subtype heterogeneity parameter $\lambda$:

$$L = (1 + \lambda)^* A + R + e \qquad (14)$$

The major subtype is defined by $\lambda = 0$, while the minor subtype assumes a heterogeneous polygenic effect induced by $\lambda \neq 0$. We varied $\lambda$ from 0 to 1 and the proportion of minor subtype cases from 0 to 0.5. Although we used a liability-threshold model, the analysis easily generalizes to the logit risk model to accommodate a stochastic structure.

**Genetic architecture models.** We modeled three distinct classes of genetic architectures to evaluate their impact on the PB-LEV correlation.

i. Polygenic genetic architecture model

In the first class, the effect size of each causal variant $i$ was assumed to be identically and independently Gaussian distributed as:

$$\beta_{\text{PB}; i} \sim \mathcal{N}\left(0, \frac{h_{\text{PB}}^2}{N_{\text{PB}}}\right) \qquad (15)$$

Under this so-called polygenic genetic architecture, each causal variant contributes a modest proportion to trait variance that depends only on the number of causal variants $N_{\text{PB}}$:

$$\text{var}(\beta_{\text{PB}; i}) := \mathbb{E}(\beta_{\text{PB}; i}^2) = \frac{h_{\text{PB}}^2}{N_{\text{PB}}} \qquad (16)$$

We assume the following linear model for the phenotype

$$y = \sum_{i=1}^{N_{\text{PB}}} G_{\text{PB}; i}\beta_{\text{PB}; i} + \varepsilon \qquad (17)$$

where the genotype $G_{\text{PB}; i}$ is standardized to have zero mean and unit variance and the residual is normally distributed as $\varepsilon \sim \mathcal{N}(\iota, \sigma_\varepsilon^\in)$. The variance of the phenotype can be written as:

$$\text{var}(y) = \frac{h_{\text{PB}}^2}{N_{\text{PB}}}ZZ^T + \sigma_\varepsilon^2 I \qquad (18)$$

where Z equals the $N \times N_{\text{PB}}$ matrix of genotype values for the $N$ samples and $\mathbf{G} = \frac{ZZ^T}{N_{\text{PB}}}$ is the genetic relationship matrix (GRM).

ii. A genetic architecture model consistent with negative selection

We considered a second class of genetic architectures, in which the minor allele frequency influences the contribution to $h_{\text{PB}}^2$ from causal variants, that is consistent with a model of negative selection (which has been shown to be present in the UK Biobank phenotypes[40]):

$$\beta_{\text{PB}; i} \sim \mathcal{N}(0, C[f_i(1 - f_i)]^{1+\alpha}) \qquad (19)$$

Here, C is a constant of proportionality that does not depend on the variant, $f_i$ is the allele frequency of the variant $i$, and, by assumption, $\alpha = -0.37$ (consistent with what was observed in the UK Biobank[40]). Using the linear model for the phenotype, we obtain:

$$\text{var}(y) = ZDZ^T + \sigma_\varepsilon^2 I \qquad (20)$$

where D is a diagonal matrix with each diagonal entry equal to C. We note that the polygenic architecture can be viewed as a subclass, with $\alpha = -1$ (and thus $C = \frac{h_{\text{PB}}^2}{N_{\text{PB}}}$).

iii. LD-adjusted kinship model

Finally, we considered the LDAK[41] class of genetic architectures:

$$\beta_{\text{PB}; i} \sim \mathcal{N}\left(0, C[f_i(1 - f_i)]^{1+\alpha}w_i\right) \qquad (21)$$

Here, the causal variant contribution to $h_{\text{PB}}^2$ is both MAF- and LD- dependent. The LD dependence is encoded in the weight $w_i$ for variant $i$, which we calculated from the European ancestry panel derived LD Score[42] $l_i$ as follows:

$$w_i = 1/(1 + l_i) \qquad (22)$$

Compared to the polygenic genetic architecture, more of the genetic signals are allocated to low LD regions. We note that the LDAK class can be viewed as a subclass of the MAF-dependent architecture that assumes an LD-weighted genotype $H_{\text{PB}; i}$:

$$H_{\text{PB}; i} = \sqrt{w_i}G_{\text{PB}; i} \qquad (23)$$

which would result in the same contribution to $h_{\text{PB}}^2$ for the causal variant. In simulations under this class, $\alpha$ was set to $-0.25$.

**Utility metric.** We define a utility metric that quantifies the usefulness of the relationship between the common-variant PB and the LEV burden in probing the genetic architecture of a trait. This metric derives from randomly sampling a set of assumed causal variants and randomly sampling the individuals from the population. In simulations, we used real genotype data (see below) to calculate the utility under different models and parameters. It is defined as the expected proportion of simulations for which, among cases, a significantly lower $(P < 0.05)$ PB in LEV carriers (i.e., individuals with the risk allele at the pathogenic variant) than in noncarriers is observed. We note that the interpretation and application of the PB-LEV test, which is performed only in cases, are fundamentally different from those of the test of an association between PB and LEV in the general population (Supplementary Information).

In order to evaluate the utility of the PB-LEV correlation for aforementioned applications for a given disease liability model ($\Lambda$) (including liability-threshold or logit risk) and a given genetic architecture ($\Gamma$) (including polygenic or "negative selection" or LDAK), we considered the correlation between LEV and PB in cases as a metric and considered its distribution. All simulations were conducted for each pair of $\Lambda$ and $\Gamma$. We systematically explored the impact of several population and disease specific parameters on this metric in simulated datasets for each such pair.

We varied several parameters of the underlying genetic architecture (Fig. 2): the heritability $h^2_{PB}$ due to the common-variant polygenic component (i.e., $h^2_{PB} = 0.1$, 0.2, 0.3, 0.4, and 0.5), the allele frequency of the LEV (i.e., $f = 1.0e\text{-}4$, 0.001, 0.005, 0.01, and 0.05), and the heritability $h^2_{LEV}$ due to the LEV (i.e., $h^2_{LEV} = 0.01$, 0.02, 0.03, 0.05, and 0.10). We varied study-level parameters, specifically the sample size $N$ (i.e., $N = 500; 1000; 2000; 3000; 5000; 10000$). Finally, we varied parameters of the model of disease liability, including the prevalence $K$ (i.e., $K = 0.001$, 0.005, 0.01, 0.02, and 0.05; the prevalence is the mean or expected disease risk probability, as shown above) and the distribution of the probability of disease risk in the population, as represented within the liability-threshold and logit risk models. Varying the parameters ($h^2_{PB}$, $h^2_{LEV}$, $f$, $N$, and $K$) within each pair ($\Lambda$ and $\Gamma$) resulted in 3750*2 simulated models for liability-threshold and logit risk model (Fig. 2). For each of the resulting models, we generated 500 simulated sets with different seeds for causal variants and performed subject sampling. In simulated cases, the Wilcoxon rank-sum test was performed to compare the PB between LEV carriers and noncarriers.

Our simulations assumed $N_{PB}$ common causal variants (MAF > 0.01) responsible for the PB of the trait. After LD pruning (PLINK command–indep-pairwise 50 5 0.01) in 23,294 ancestrally European samples (see below), we set $N_{PB}$ equal to 95,593. The effect size of common variants was defined based on different genetic architectures (see above). The effect size of the LEV, with allele frequency $f$, was defined as follows:

$$\beta_{LEV} = \sqrt{\frac{h^2_{LEV} var(y)}{2f(1-f)}} \qquad (24)$$

The phenotype (y) was then simulated as follows:

$$y = \sum_{i=1}^{N_{PB}} G_{PB; i}\beta_{PB; i} + G_{LEV}\beta_{LEV} + \varepsilon \qquad (25)$$

$$\varepsilon \sim \mathcal{N}(0, 1 - h^2_{PB} - h^2_{LEV}) \qquad (26)$$

Here $G_{PB; i}$ and $G_{LEV}$ are the genotypes for the common variant $i$ and the LEV, respectively. The genotypes $G_{PB; i}$ were standardized to have zero mean and unit variance. The total genetic risk $x$ was mapped to a disease risk probability $p = \Phi^*(x)$ or $p = 1/(1 + e^{-(x+u_0)})$, under a liability-threshold or a logit risk, respectively.

**LEV heterogeneity**. A more general scenario assumes potential LEV heterogeneity, i.e., at least one LEV – indexed by, say, $m$ – each with effect size $\beta_{LEV,m}$ and allele frequency $f_m$. For example, different carriers may carry different LEVs. Here the heritability due to these variants (of count $N_{LEV}$ in the population) is given by:

$$h^2_{LEV} = 2\sum_{m=1}^{N_{LEV}} (\beta_{LEV,m})^2 f_m(1 - f_m)/var(y) \qquad (27)$$

In this case, our simulation framework, as a simplifying assumption, uses the LEV with the largest causal contribution to $h^2_{LEV}$. That is, for simulations, we chose:

$$(\beta_{LEV}, f) = \underset{m}{\text{argmax}}(\beta_{LEV,m})^2 f_m(1 - f_m) \qquad (28)$$

Thus, under the assumption of LEV heterogeneity, we are using a lower bound estimate for $h^2_{LEV}$ (equal to the contribution of the LEV with the largest contribution to $h^2_{LEV}$).

**Empirical data informed simulations**. In order to capture the LD structure and real genomic data, simulations were performed using the BioVU[43] European ancestry subjects (23,294 independent individuals). After quality control, common small-effect variants (MAF > 0.01) that passed an imputation quality cutoff ($R^2 > 0.30$) were included in the PB in downstream analyses. LEVs (whose MAF ranged from 1e-4 to 0.05) were used in the simulations. We also performed the sampling of subjects in our simulations from this dataset.

**Statistical power**. Since the PB is typically estimated using a PGS without knowledge of which variants are causal, we simulated models with a certain pro-portion, $\pi_1$, of variants that comprise the PGS assumed to be causal. $\pi_0 = 1 - \pi_1$ is then the proportion of null (noncausal) variants in the estimate PGS of PB. For each distribution $D_j$ of causal effects as defined above (representing a class of genetic architectures; $j = a, b, $ or $c$), we defined a mixture distribution to generate a PGS consisting of both causal and noncausal variants:

$$\beta_{PB; i} \sim T_j = \pi_1 D_j + \pi_0 \delta_0 \qquad (29)$$

where $\delta_0$ is the point-mass distribution. (For our purposes, $\pi_0 \in \{0.1, 0.3, 0.5, 0.7, 0.9\}$).

The power was then calculated as the proportion of simulations for which, among cases, a significantly lower ($P < 0.05$) PGS in LEV carriers than in noncarriers is observed, assuming a certain proportion of noncausal variants in the PGS.

**Testing the PB-LEV correlation in real datasets**. We tested several complex traits and included previously published and available genomic data from large case–control analyses of TS, OCD, and T1D in our analyses (Supplementary Table 1)[7,30,44–50]. We calculated the common-variant PGS for each individual using the genomic best linear unbiased prediction (GBLUP) method implemented in GCTA[51], estimating the total genetic effect for each individual from the SNPs used to estimate the GRM. For the LEVs of neuropsychiatric disorders (i.e., TS and OCD), we restricted analysis to the pool of rare variants most likely to be enriched for pathogenic variation, including copy number variants (CNVs) that are large (>500 Kb), rare (<1% in the Database of Genomic Variation), and genic (con-taining at least one gene), and LoF or putative damaging (missense) single nucleotide variants identified in previous reports[7,44,50] (Supplementary Table 2). For T1D, we used the *DRB1* susceptibility locus[52] of the HLA region which accounts for ~50% of the heritability for T1D[53] (Supplementary Table 2). The HLA-*DRB1* genotype data was obtained from the WTCCC with permission for analysis. We compared PGS between cases with and without LEVs using Wilcoxon rank-sum test.

**Application to large-scale biobank**. Here, we illustrate the framework in the context of empirical data, i.e., with empirically observed values of heritability, disease prevalence, and LEV frequency for traits in the UK Biobank (involving up to 361,194 samples), to (1) demonstrate that the predictions and findings generated based on the models of the framework recapitulate empirical results and (2) show real-world applications of the framework. The common variant-based (MAF ≥ 0.01) heritability was obtained from the Ben Neale lab (http://www.nealelab.is/uk-biobank/). In total, 2322 traits were included after removing nonsignificant ($P_{h^2} > 0.05$) or low-heritability ($h^2 < 0.05$) traits. We leveraged the effect size (OR) and allele frequency for the LEVs, as presented in a recent exome-wide phenome-wide association study in the UK Biobank[27]. Large-effect (OR ≥ 1.5) trait-associated non-synonymous variants under the allelic model were considered as LEVs. We also extended these analyses to another set of four traits, including breast cancer, colorectal cancer, type 2 diabetes, and short stature[28] (We excluded osteoporosis from the study since its estimated OR was not significant). For each tested trait, we used the top-ranked LEV in the downstream analysis. In total, 36 heritable traits with at least one LEV were included in additional analyses (Sup-plementary Data 2).

The heritability attributable to the LEV was estimated from the "case status by LEV status" table (if available) or from the OR estimate from logistic regression using an interpretable measure of $R^2$ (coefficient of determination)[54]. Its expected value $\mathbb{E}(R^2)$ equals the heritability on the liability scale. Under the probit-liability scale, the estimator is calculated as follows:

$$R^2_{probit} = \frac{var(\ln(\hat{OR})g)}{var(\ln(\hat{OR})g) + 1} \qquad (30)$$

Under the logit-liability scale, the estimator is defined as follows:

$$R^2_{logit} = \frac{var(\ln(\hat{OR})g)}{var(\ln(\hat{OR})g) + 3.29} \qquad (31)$$

The genotype dataset was simulated using PLINK[55] with the sample size matched with the UK Biobank (Supplementary Data 2). We used the distribution of MAF and the LD score[56] as calculated from real genomic data (in 23,294 European ancestry samples).

We calculated the utility of the PB-LEV correlation, using the empirically-informed parameters for the tested traits in the UK Biobank. In addition, we were interested in determining, for a given PB profile, the probability of LEV carriers in cases, $P(LEV \mid Y = 1)$, which we calculated using the "number of LEV carriers per 1000 cases". This fraction quantifies the effectiveness of our PB-based approach for prioritizing cases for sequencing —a practical application. For a given trait, cases were grouped into ten equally-sized bins according to their PB risk (i.e., polygenic risk score). The number of LEV carriers per 1000 cases was calculated for samples in each of the bins. The resulting proportion was then tested against empirical observations, specifically for the set of four traits (breast cancer, colorectal cancer, type 2 diabetes, and short stature), for which data were publicly available for the comparison. For the simulations in the UK Biobank data, we assumed the genetic architecture consistent with negative selection, which had been inferred from the UK Biobank[40]. Both the liability-threshold and logit risk models were considered.

**Reporting Summary**. Further information on research design is available in the Nature Research Reporting Summary linked to this article.

## Data availability
The summary-statistics on the UK Biobank are publicly available. The download link and the technical details can be found at http://www.nealelab.is/uk-biobank and http://www.nealelab.is/blog/2017/9/14/heritability-501-ldsr-based-h2-in-ukbb-for-the-technically-minded. The trait-associated variants from exome sequencing come from recent published studies [https://doi.org/10.1101/2020.12.13.422582][https://doi.org/10.1038/s41436-020-01007-7][27,28]. The BioVU summary-statistics, including the MAF and LD score data, can be retrieved from https://zenodo.org/record/4767933. All requests for raw

BioVU data (for example, genotype) are reviewed by Vanderbilt University Medical Center to determine whether the request is subject to any intellectual property or confidentiality obligations. For example, patient-related data not included in the paper may be subject to patient confidentiality. Any such data and materials that can be shared will be released via a material transfer agreement. The simulation studies were performed using the software we created for this project (https://github.com/gamazonlab/Polygenic_Background_Rare_Variant_Axis). These simulations were informed by linkage disequilibrium patterns and allele frequency information from empirical data (BioVU) and the simulation parameters reflected empirical parameters from disease phenotype data (UK Biobank). Information on TS and OCD is available from the published GWAS of TS[30,31,45] and OCD[30,46]. GWAS summary-statistics and data access for TS and OCD can be obtained from the Psychiatric Genomics Consortium website: https://www.med.unc.edu/pgc/pgc-workgroups/ocd-tourette-syndrome/. We analyzed the previously published WTCCC T1D dataset. WTCCC data access for T1D cases and controls is described at https://www.wtccc.org.uk/info/access_to_data_samples.html. Source data are provided with this paper.

## Code availability

The source code can be downloaded from https://github.com/gamazonlab/Polygenic_Background_Rare_Variant_Axis.

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

## Acknowledgements

E.R.G. is grateful to the President and Fellows of Clare Hall, University of Cambridge for providing a stimulating intellectual home and for the generous support. This research is supported by the National Institutes of Health (NIH) Genomic Innovator Award R35HG010718 (E.R.G.), NIH/NHGRI R01HG011138 (E.R.G.), and NIH/NIGMS R01GM140287 (E.R.G.). L.K.D. was supported by grants from the National Institutes of Health, including R01NS102371, R01MH113362, R01MH118223, R01NS105746, and R56MH120736. E.R.G., L.K.D., and D.Z. are thankful to Naomi Wray for critical reading of the manuscript and her feedback. Initial support for the Synthetic Derivative was provided by the National Center for Research Resources, Grant UL1 RR024975-01, and is now at the National Center for Advancing Translational Sciences, Grant 2 UL1 TR000445-06. The content is solely the responsibility of the authors and does not necessarily represent the official views of the NIH. The datasets used for the analyses include Vanderbilt University Medical Center's BioVU, which is supported by numerous sources: institutional funding, private agencies, and federal grants. These include the NIH-funded Shared Instrumentation Grant S10RR025141; and CTSA grants UL1TR002243, UL1TR000445, and UL1RR024975. Genomic data are also supported by investigator-led projects that include U01HG004798, R01NS032830, RC2GM092618, P50GM115305, U01HG006378, U19HL065962, R01HD074711; and additional funding sources listed at https://victr.vumc.org/biovu-funding/. All research including BioVU data was deemed non-human subjects research by the VUMC IRB (IRB#190418, 172020, and 160302).

## Author contributions

E.R.G., L.K.D., and D.Z. designed the study and wrote the manuscript. E.R.G. and D.Z. designed the simulation framework and the application to biobank data. D.Z. performed the simulations. D.Z. and L.K.D. conducted the analysis of the empirical datasets. S.H.L., D.Y., J.M.S., C.A.M., L.M.G., and E.C. provided critical input and contributed to the review and editing of the manuscript. E.R.G. directed and acquired funding for the study.

## Competing interests

E.R.G. receives an honorarium from the journal *Circulation Research* of the American Heart Association, as a member of the Editorial Board. The remaining authors declare no competing interests.
