## [Peer Review File · Nature Communications]

REVIEWER COMMENTS

Reviewer #1 (Remarks to the Author):

The paper by Zhou et al uses simulation to assess what information can be gained from considering both common and rare variants in the context of risk prediction. The authors use different models and genetic architecture, mainly to assess the correlation between polygenic burden and carrier status for high impact mutations.

Developing is framework is original and could potentially be very useful. However, I have several concerns about this work. The authors run a very large number of different simulations, varying many parameters and using different models. They have only partially worked out the key messages that this work provides.

Detailed comments

The abstract does not really contain any results. Clearly it is hard to distil them to a few sentences in this complex setting but it also reflects on the fact that the authors have not quite worked out the conclusions.

The background is very nicely written although it was not clear to me why they discussed collider bias and spurious associations in case only settings at this point (seemed like a half-motivated response to a reviewer comment).

One of the assumptions is that the high penetrance mutations and PRS are independent and additive. The authors should justify this. I can see arguments that they might not always be, e.g. affecting proteins in the same pathway might not be an independent effect. It would be interesting to assess this explicitly using their framework.

The abbreviations are terrible. Throughout it took a lot of mental resources to just remind myself what these things are – LEV low effect variants? Pebvs what? There must be a better way to describe high impact mutations and polygenic effects.

It would be helpful if the authors explain the differences between the two models they use, liability threshold and logistic regression, in terms of what the different assumptions are how disease risk and disease status relate.

The section on utility if polygenic – rare variant correlation should be the heart of the paper. However, I am struggling to take any useful insights out of this. There are not many concrete findings and those that are reported are trivial, for example that utility is higher when genetics explain more variance.

What is missing is the answer to the question, what is the utility of the polygenic-rare variant correlation. I would suggest improving this in two ways. Firstly, the authors could try to define a utility threshold that would be impactful. Secondly, they could match their findings against real life diseases. For many of them the relevant parameters, such as polygenicity and var explained are known. So what is the utility for BC, T1D, hypertension etc. They could do this based on current performance of PRS and potential performance given the h^2 .

Why would and LEV allele frequency of $LEV < 0.005$ be an extreme case. In fact, I suspect that most LEV's are that rare.

The test whether PRS are as predictive as LEV's is superficial and very unclear. This is a highly disputed matter and requires a more dedicated assessment. This includes consideration (an mention) what PRS percentiles are compared and what realistic scenarios for the rare and common h^2 are.

I find the conclusions much too strong given the results: "our study suggests that cSEV-PB may be a useful addition to the prioritization schemes of samples likely to harbour rare pathogenic variants". How was this demonstrated? For what diseases and under what architectures?

Reviewer #2 (Remarks to the Author):

The manuscript by Zhou et al. describes simulation studies of the correlation of rare large effect variants and common variant (polygenic) background in disease using different genetic architectures, disease liability models, and study parameters. The simulations provide a quantification of power and utility by varying assumptions of study parameters that are disease and sample specific. They then provide a brief analysis on empirical data on three diseases from recent studies. Overall, the study provides important methodological insight into a clinically important topic that has been subject of multiple recent publications on monogenic-polygenic interaction and disease penetrance across a variety of diseases. The manuscript is well written for the most part, particularly as it details the importance of the subject with regards to clinical translation and could help investigators studying the subject on empirical data for specific disease as they think about the study design and analysis. I hope the below comments could help the authors improve the manuscript prior to publication.

1- While the background, importance and hypothesis are clearly described upfront, the reader struggles to understand in brief what are the methods. While the different parameters and models used for the simulations are made clear, some important details regarding samples are left from the abstract. On page 5, there is discussion of case-only analysis, but it is not clear how that factors into this study. Also, it is not until page 14 that the 23,294 participants from BioVu are mentioned and it is not clear how those are used. For the empirical analysis on the three diseases, important sample details in supp tables 1-3 should be described in the main manuscript of the text.

2- Additionally, the punchline findings or conclusions of the study are not stated upfront and one has to read through to distinguish what is background and rationale and what is actual findings of the study. It would be helpful at the end of the conclusion section to state the main findings of the study in a list format.

3- While the bulk of the manuscript is describing simulations, it would be helpful to provide results from empirical analysis on more than the three disease examples, particularly to illustrate whether some of the hypotheses generated based on simulations do actually hold in the context of empirical data for diseases with variable heritability, diseases prevalence and LEV frequency.

4- The authors make the argument for reduced polygenic background effect among LEV carriers vs. non-carriers as a generalizable finding based on one recently published example for breast cancer, but it is still not clear that this is generalizable across diseases and

studies. It was not seen in OCD in this study, and more notably it was not seen in other recently published studies other than the one they cite. Eg. Kuchenbaecker et al JNCI 2017, Oetgens et al. Nat Comm 2019 and Fahed et al Nat Comm 2020). Although often limited by statistical power, there was no evidence of interaction between LEV and polygenic background in those studies, and significant stratification of risk by polygenic background among LEV carriers. It is helpful to discuss the simulations in this study in context of some clinically important disease examples such as those recently reported.

5- The discussion on how a low polygenic background risk can explain reduced penetrance of LEV carriers under the liability threshold model is important. Recent study by Fahed et al Nat Comm 2020 highlighted this concept across three diseases. Authors comment (last paragraph page 24) that their findings contrast this study, but it is not clear how that is the case, as the comment seems to be consistent with this study.

6- Minor comment: references 23 and 49 are duplicated

Reviewer #3 (Remarks to the Author):

Zhou and co-workers create an extensive simulation framework, partly based on real-world data (LP-structure, negative selection constants, known large effect variants, etc) and partly on simulated varied inputs of small and large effect heritabilities, frequencies, proportion of non-causal variants as well as a setup of different genetic architecture models.

It is noted that the simulation framework predictions are concordant with several observations that seems to be well-established, even self-evident. These include: that utility decrease with decreased proportion of causal variants, that it decreases with lower HSNP2 both for common and rare variants. However, several less self-evident predictions are made in the model and these are the key novelty of the manuscript.

I have the following comments to the manuscript:

1

The utility metric is a central part of the first two pages of results and figure 3. According to methods, it is defined as "the expected proportion of simulations for which, among cases, a significantly lower ($P < 0.05$) cSEV-PB in LEV carriers than in non-carriers is observed" (from methods). It took me several reads to understand this. That there is less cSEV-PB in LEV carriers than non-carriers. I if understand correct. And I'm still unsure why this is called utility. Utility towards what? I think it would improve the manuscript to be clearer about what it is, and why it is important - also in the results section (right now it only back-refers to methods).

2

Because this frame-work could form a standard in future rare and common variant combinations, perhaps you should consider a better and simpler acronym than "cSEV-PB". I have no specific suggestions but found the current acronym very clunky.

3

A number of very concrete testable hypotheses are posed, but very few of them are tested in application to real world data. Principally the real-world testing only briefly describes testing of the polygenic burden in tourettes and obsessive-compulsive disorder, with or without CNVs. The manuscript would benefit from a clear listing of what other hypotheses are

recommendations for future studies.

4

In supplementary figure 10 there's discrepancy between the caption text and the simulated constants printed above the figure, e.g. in figure a, the $h^2_{LEV}=0.03$, but in caption it says $h^2_{rare}=0.3, h^2_{rare}=0.03$). Also, it seems that only two different combinations are simulated (according to caption).

5

It is not very easy to reproduce the findings from the code in the github repository. There are many external links (summary `<-read.table(paste0('~/../Dropbox/POLYGENIC_LARGE_EFFECT/simulation/results/result_)`) and un-stated dependencies, e.g. `cmd=paste0('/home/zhoud2/tools/plink2/plink2 -bfile /data/c***/z***/projects/prslev/common/common_all --score)`. I think this could be improved, as I at least could not reproduce the simulations even after much tinkering.

Reviewer #4 (Remarks to the Author):

In this paper, Zhou et al. presented a simulation study to assess the power to detect a difference in cSEV-PB among cases between LEV carriers and non-carriers. The main clinical implication is that subjects with lower polygenic burden may be at higher risk of a carrier of a large-effect rare variant. The simulations are comprehensive and sound, here are my concerns and suggestions.

1) A key assumption is that common and rare variants are independent and additive. Could the authors discuss this further, eg possibility that it could be violated? For example, common variants can act as effect modifiers of rare mutations.

2) Are there ways to link up simulation parameters with real gwas/genomics data? Eg using heritability, N_{poly} and distribution of effect size of variants as derived from some real data? This could be difficult I suppose but I would suggest also briefly discuss it as a limitation (ie difficulty in choosing parameters that fully match actual diseases).

3) the utility metric is quite key to the assessment. I may have missed it, but do authors compared PRS directly between cases and non-cases, like by a t-test or something like that?

'It (Utility) is defined as the expected proportion of simulations for which, among cases, a significantly lower ($P < 0.05$) cSEV-PB in LEV carriers than in non-carriers is observed'. I wonder if we can quantify how much lower the common variant burden is (eg a min. threshold can be set? Or other methods). For instance, 3% lower cSEV-PB may be significant when sample size is large, but may not be clinically relevant. I am not sure how best to quantify the utility measure, but I feel that just measuring the proportion of (simulation) cases with sig. higher burden is inadequate.

Therefore I also feel it is still hard to say whether cSEV-PB will be clinically useful yet, as patients can still have similar polygenic burden (PB) but how well can we discriminate carriers and non-carriers?

4) The authors have done extensive simulations, however given our limited understanding of the genetics of most diseases, there are a number of important limitations, and I suggest greater caution when proposing the applicability of the finding in clinical practice.

I therefore suggest to provide a more thorough discussions of the limitations of the simulation approach and studies conducted, eg

- Simulations are useful but it is still hard to mimic real disease genetics , eg no, of causal variants, genetic architecture, normal distribution of effect sizes are very hard to verify
- Assumptions of independence (and additivity) between common and rare variants unlikely to be true in many cases
- Remains to be studied whether the findings are generalizable, given varying genetic architecture of diff diseases
- For OCD (or TS), I suppose the sample size is still not very large and the roles of large-effect rare variants are still under study, so one may not be able to show the relationship between PB and LEV

REVIEWER COMMENTS

Reviewer #1 (Remarks to the Author):

The paper by Zhou et al uses simulation to assess what information can be gained from considering both common and rare variants in the context of risk prediction. The authors use different models and genetic architecture, mainly to assess the correlation between polygenic burden and carrier status for high impact mutations.

Developing this framework is original and could potentially be very useful. However, I have several concerns about this work. The authors run a very large number of different simulations, varying many parameters and using different models. They have only partially worked out the key messages that this work provides.

RESPONSE: We thank the reviewer for the constructive feedback, which has substantially improved the manuscript. We agree with the reviewer that our study will potentially be very useful. In the revised manuscript, we have conducted new analyses, present new results, and (we believe) emphasized the key messages much more effectively. For example, we applied our **summary-statistics-based framework** using parameters empirically extracted from the UK Biobank and have highlighted the key messages from the framework. In the revised manuscript, we applied the framework to 36 traits with available LEV data (Supplementary Table 3) using empirically-informed values (i.e., derived from the UK Biobank) for the simulation parameters (including heritability, disease prevalence, and LEV frequency). The fact that the framework is built on summary statistics will be extremely useful to the scientific community because this will facilitate analysis of publicly available datasets, as we illustrate in the revised paper. The concordance of the predictions based on the models of the framework and empirical data provides strong support to the methodology, but also allowed us to highlight the applications to which the framework can be put. We provide software (PB-LEV-SCAN, available on GitHub for download) to enable these applications. We added a conclusion section (also included below) at the end of the revised manuscript summarizing the key messages from the study. Please see also our detailed responses, including the specific changes we have made to the manuscript in response to the reviewer's concerns. We have added a new concluding paragraph:

In conclusion, we developed a summary-statistics-based framework that utilizes the relationship between PB and LEV among cases for some methodologically- and clinically-relevant applications, including PB-based LEV screening. The framework's testable, falsifiable predictions on the proportion of LEV carriers among cases with a given PB profile under pre-specified simulation parameters were confirmed by empirical data.

Application of the framework to a large-scale biobank (UK Biobank) showed that the effectiveness of PB-based LEV screening varied substantially by trait. Taken together, these findings shed critical light on the use of PB in clinical practice.

Detailed comments

1. The abstract does not really contain any results. Clearly it is hard to distil them to a few sentences in this complex setting but it also reflects on the fact that the authors have not quite worked out the conclusions.

RESPONSE: We thank the reviewer for the comment. We agree that it is hard to summarize the results from the whole framework. We made some edits (included below) to the abstract which now highlights the key messages and some of the primary results.

Studies of the genetic basis of complex traits have demonstrated a substantial role for common, small-effect variant polygenic burden (PB) as well as large-effect variants (LEV, primarily rare). Here, using the two sources of genetic risk among individuals who share a diagnosis, we investigated how the genetic architecture and disease-liability model influence risk stratification and prediction. We identified sufficient conditions in which GWAS-derived PB may be used for well-powered rare pathogenic variant discovery or as a sample prioritization tool for whole-genome or exome sequencing. Through extensive simulations of genetic architectures and generative models of disease liability with parameters informed by empirical data (BioVU, n=23,294; UK Biobank, n up to 361,194), we quantified the power to detect, among cases, a lower PB in LEV carriers than in non-carriers. The resulting summary-statistics-based methodology (with publicly available software, PB-LEV-SCAN) makes predictions on PB-based LEV screening for 36 complex traits, which we confirmed in several disease datasets with available LEV information in the UK Biobank. Our results uncovered clinically useful conditions wherein the risk derived from the PB is comparable to the LEV-derived risk and underscore the ways PB may modify rare variant penetrance. Finally, we found that the influence of negative selection on genetic architecture could be observed through its greater effect on the PB-LEV correlation in comparison with the neutral polygenic architecture. These findings have implications for the use of polygenic risk scores in clinical decision-making and may prove broadly useful for the design of future genetic studies across the human disease phenome.

2. The background is very nicely written although it was not clear to me why they discussed collider bias and spurious associations in case only settings at this point (seemed like a half-motivated response to a reviewer comment).

RESPONSE: The inclusion of collider bias in the Introduction was to clarify that the PB-LEV (polygenic burden and large-effect variant) correlation among cases is fundamentally different from the test of association between PB and LEV in the general population. We have moved the paragraph on the collider bias to Supplementary Information (included below). We have made the clarification in the following sections:

(Methods)

... We note that the interpretation and application of the PB-LEV test, which is performed only in cases, are fundamentally different from those of the test of an association between PB and LEV in the general population (Supplementary Information).

Notes for case-only study design

We compared the polygenic burden (PB) between large-effect variant (LEV) carriers and non-carriers among cases. We note that the interpretation and application of the PB-LEV test, which is performed only in cases, are fundamentally different from those of the test of an association between PB and LEV in the general population. A lower PB among LEV carriers than non-carriers could not be interpreted as a negative correlation between PB and LEV in the general population (a consequence of collider bias). In fact, PB and LEV are not correlated in the general population by study design. Generally speaking, collider bias can result in biased genetic associations or bias in associations between variables that influence study participation and should be avoided^{1,2}. Spurious associations (e.g., genetic associations or associations between variables) can arise in the absence of true correlation in the intended study population³. However, in this study, our interest is elsewhere (i.e., not in inferring the correlation between PB and LEV in the general population).

3. One of the assumptions is that the high penetrance mutations and PRS are independent and additive. The authors should justify this. I can see arguments that they might not always be, e.g. affecting proteins in the same pathway might not be an independent effect. It would be interesting to assess this explicitly using their framework.

RESPONSE: We thank the reviewer for these critical points. Several recent empirical studies reported that trait-associated common, small effect variants and large-effect variants were independent and their risk burdens were additive^{4,5,6}. However, as the reviewer noted, it is possible that independence and additivity do not hold for some complex traits.

Therefore, we performed simulations of two additional scenarios wherein (1) small-effect common variants and LEV are dependent and (2) the interaction between small-

effect common variants and LEV also contributes to disease risk. We have added the following new results and a new Figure 6 (included below).

For the first scenario of dependence, we assumed that the LEV was in high LD ($D'=1$) with the common variant with the largest effect size. We performed simulations and compared the utility of the PB-LEV correlation with the utility when PB and LEV were independent. For the simulations, we set $h_{PB}^2 = 0.3$, $h_{LEV}^2 = 0.01$, $f = 0.001$, and prevalence (K) = 0.01, and assumed a polygenic genetic architecture and the liability-threshold model. The values for the utility were similar for the dependent and independent cases as the sample size was varied from 500 to 10,000 (Figure 6a).

For the second scenario of interaction, we assumed that the interaction between the LEV and the common variant with the largest effect size contributed to the trait variance ($h_{interaction}^2 = 0.02$). For all other parameters, we used the same settings as above. Here again, the utility values between the disease models with and without interaction were similar (Figure 6b).

Figure 6. Robustness to independence and additivity of PB and LEV. (a) To test the effect of potentially dependent PB and LEV (in the general population) on the PB-LEV correlation (in cases), we assumed that the LEV was in high LD ($D'=1$) with the top-ranked common variant (i.e., with the largest effect size) and compared the utility. Here, we set $h_{PB}^2 = 0.3$, $h_{LEV}^2 = 0.01$, $f = 0.001$, and prevalence (K) = 0.01, and assumed a polygenic genetic architecture and the liability-threshold model in the simulations. (b) To test the extent to which potential interactions between the PB and LEV could affect the PB-LEV correlation, we assumed that the interaction between the LEV and the

common variant with the largest effect size contributed to the variance of trait ($h_{interaction}^2 = 0.02$) and performed simulations. All other parameters were fixed as in (a).

4. The abbreviations are terrible. Throughout it took a lot of mental resources to just remind myself what these things are – LEV low effect variants? Pebvs what? There must be a better way to describe high impact mutations and polygenic effects.

RESPONSE: We thank the reviewer for the comment. We have now simplified the “cSEV-PB” (short for common, small effect variant polygenic burden) to “PB”. We have clarified the definition in the revised manuscript without sacrificing the accuracy. To attain a balance of accuracy and simplicity, we kept the “LEV” to denote the “large-effect variant”.

5. It would be helpful if the authors explain the differences between the two models they use, liability threshold and logistic regression, in terms of what the different assumptions are how disease risk and disease status relate.

RESPONSE: We thank the reviewer for the opportunity to clarify the differences between the liability-threshold model and the logit risk model. As we mentioned in Methods, the genetic risk score x_{logit} under the logit risk model maps to the same disease risk probability (i.e., $p = p_{logit} = p_{liability-threshold}$). The major difference between the liability-threshold model and the logit risk model is that the former one defines cases by a liability threshold (so that cases are deterministically defined) and the latter one defines disease status by sampling from a binomial distribution given a probability of disease (so that the definition of a case has a stochastic structure). We have now added the necessary clarification on page 22 of Results (**all the page numbers in our responses refer to the revised version of the manuscript with TRACK CHANGES**).

Additionally, in the updated simulation script (github.com/gamazonlab/Polygenic_Background_Rare_Variant_Axis), we provided options for these two disease risk models for users. By comparing the predictions from our summary-statistics-based framework with empirical observations in the UK Biobank (four new traits: breast cancer, colorectal cancer, type 2 diabetes, and short stature, which are available for this comparison⁷; details are included in the response to comment #7), we found that both the liability-threshold model and the logit risk model fit empirical data well (Figure 10 and Supplementary Figure 15).

6. The section on utility if polygenic – rare variant correlation should be the heart of the paper. However, I am struggling to take any useful insights out of this. There are not many concrete findings and those that are reported are trivial, for example that utility is higher when genetics explain more variance.

RESPONSE: We thank the reviewer for the valuable comments. We considered the *utility* as an intuitive metric to quantify the usefulness of the relationship between the common-variant polygenic burden and the LEV burden in probing the genetic architecture of a trait. We have now performed new analyses and illustrated the use of our summary-statistics-based framework for potential (research or clinical) applications. The framework would predict that a low-PB case should have a higher probability of being an LEV carrier. In the revised manuscript, we have included a new metric from the simulations, i.e., the probability of an LEV carrier given case status $P(LEV | Y = 1)$, which we calculated using the ‘number of LEV carriers per 1000 cases’, for a given PB profile, to make predictions on PB-based LEV screening in empirical datasets with parameters matching the parameters of the simulations. Notably, we confirmed the predictions using empirical observations from the UK Biobank (please find more details in the response to comment #7).

7. What is missing is the answer to the question, what is the utility of the polygenic-rare variant correlation. I would suggest improving this in two ways. Firstly, the authors could try to define a utility threshold that would be impactful. Secondly, they could match their findings against real life diseases. For many of them the relevant parameters, such as polygenicity and var explained are known. So what is the utility for BC, T1D, hypertension etc. They could do this based on current performance of PRS and potential performance given the h^2 .

RESPONSE: We thank the reviewer for the suggestions. We now use 0.80 as the threshold for utility and power. A dashed horizontal line at 0.80 was added for each of the relevant figures. We also applied the threshold in additional empirical data-based analyses. Leveraging the availability of parameters (including heritability, MAF and effect size of LEV, and prevalence) derived from the UK Biobank, we have now applied our framework to 36 empirical traits with available LEV data (Supplementary Table 3) and estimated the utility for each trait. The common-variant-based heritability had been derived (Neale Lab) from chip-based common SNPs ($MAF \geq 0.01$). The effect size and MAF of the LEV were derived from a variant-level Exome-wide association study on the UK Biobank samples⁸. The heritability attributable to the LEV was estimated from the “case status by LEV status” table (if available) or from the OR estimate from logistic regression using an interpretable measure of R^2 (coefficient of determination, see Methods on page 20 under the section *Application to large-scale biobank*)⁹. Leveraging the trait-specific empirically-informed parameters, we applied our framework and

determined the probability of an LEV carrier given case status $P(LEV | Y = 1)$, which we calculated using the ‘number of LEV carriers per 1000 cases, for different levels of PB. As we showed in Figure 9 (included below) and Supplementary Figure 16, a low-PB case has a higher probability of being an LEV carrier, indicating that PB is very informative for prioritizing samples for LEV screening.

To demonstrate the value of our summary-statistics-based framework, we have applied it to a large-scale biobank (UK Biobank) involving 36 traits with available LEV data (Supplementary Table 3). As an example (Figure 9a), we would expect to find 27 cases carrying a stop-gain mutation (OR = 5.1, MAF = 0.0013) on *MYOC* per 1000 glaucoma cases from the lowest PB risk bin. However, among the 1000 glaucoma cases in the highest PB risk bin, we would expect to find only 6 carriers. We analyzed four additional traits (Figure 9b, 9c, 9d, and 9e) representing three distinct patterns. The results on malignant melanoma (Figure 9b) indicated that only cases with low PB risk would be worth sequencing for LEV screening. For both Crohn’s disease (Figure 9c) and acute tonsillitis (Figure 9d), the striking difference in the estimated proportion of LEV carriers among cases from the different PB risk bins would suggest the usefulness of PB-based prioritization. Per 1000 Crohn’s disease patients, 214 cases would be expected to be *NOD2* frameshift mutation carriers from the lowest PB risk group whereas only about 67 cases from the highest PB risk group would be expected to be carriers of the mutation (Figure 9c). Nearly half of the acute tonsillitis cases (452 / 1000) from the lowest PB risk group were predicted to be *OXCT2* missense mutation carriers, which was a nearly 3-fold increase relative to the highest PB risk group (Figure 9d). However, PB-based prioritization appears to be less helpful for screening low frequency, large-effect type 2 diabetes-associated variants (Figure 9e).

Moreover, **we confirmed the framework’s predictions by comparison with empirical observations for four complex traits**, namely, breast cancer, colorectal cancer, type 2 diabetes, and short stature, which are available for this comparison⁷. As we showed in Figure 10 (included below), the empirical data (i.e., the actual proportion of LEV carriers, marked as a red diamond) were highly concordant with our framework’s predictions (shown in boxplot).

Figure 9. Cases with low polygenic risk score have higher probability of carrying an LEV. Based on empirically-informed parameter values (i.e., derived from the UK Biobank), we performed simulations (under the liability-threshold model and the genetic architecture in line with negative selection) and compared the number of LEV carriers per 1000 cases among the different polygenic risk scores for (a) glaucoma, (b)

malignant melanoma, (c) Crohn’s disease, (d) acute tonsillitis, and (e) type 2 diabetes. For each trait, we grouped the cases into 10 equally-sized polygenic risk score bins. For each bin, the mean \pm 1sd of the number of LEV carriers per 1000 cases is displayed in blue circles and bars. The distribution of polygenic risk score is shown as a histogram. For example, per 1000 Crohn’s disease cases, 214 would be expected to be *NOD2* frameshift mutation carriers from the lowest PB risk group; in contrast, PB-based LEV screening would be less effective for type 2 diabetes. The proportion of LEV carriers in the lowest PB risk group for acute tonsillitis was a nearly 3-fold increase relative to the highest PB risk group. Thus, the framework (assuming pre-specified simulation parameters) provides testable predictions on the number of LEV carriers per 1000 cases with a given PB profile and on the sample polygenic risk score profile to optimize LEV screening.

Figure 10. Prediction of the framework on proportion of LEV carriers in cases with a given PB profile matches empirical observations, providing an LEV screening approach. Using empirically-informed values (i.e., derived from the UK Biobank) of the parameters (including the common variant based heritability; disease prevalence; allele frequency and odds ratio of the LEV), we performed simulations for four traits under the genetic architecture in line with negative selection. We varied the π_0 (the proportion of non-causal variants in the polygenic risk score) at (a) 0, (b) 0.5, and (c) 0.8. Cases were defined under the liability-threshold model and classified into ‘low-PB’ and ‘high-PB’ (using the median as the cutoff) groups. The proportion of LEV carriers in each group was defined as the number of cases carrying LEV over the total number of cases. The distribution of the proportion (across 500 simulated sets) is shown in the boxplot. The

median of the proportion is visualized as a black segment in the middle of the box. The lower and upper hinges correspond to the first and third quartiles (the 25th and 75th percentiles). The upper / lower whisker extends from the hinge to the largest / smallest value no further than / at most $1.5 * \text{IQR}$ from the hinge (where IQR is the inter-quartile range or the distance between the first and third quartiles). The actual observed proportion of LEV carriers for each PB profile from empirical data (with the matching parameters as the simulations) are marked as a red diamond. Thus, the prediction of the framework and the empirical dataset (with matching parameters as the simulations) were concordant.

8. Why would and LEV allele frequency of $\text{LEV} < 0.005$ be an extreme case. In fact, I suspect that most LEV's are that rare.

RESPONSE: We agree with the reviewer. We removed the word “extreme” on page 24.

9. The test whether PRS are as predictive as LEV's is superficial and very unclear. This is a highly disputed matter and requires a more dedicated assessment. This includes consideration (an mention) what PRS percentiles are compared and what realistic scenarios for the rare and common h^2 are.

RESPONSE: We thank the reviewer for the valuable comments. We have now clarified what PRS percentiles we did use and have summarized the OR comparison results. In the revised manuscript, we emphasize that our framework is based on realistic scenarios, i.e., from parameters (LEV effect size and common h^2) based on empirical datasets, i.e., BioVU and the UK Biobank.

We have added the following to Methods (see “Application to large-scale biobank” on page 19-20), to highlight that the simulation parameters were based on empirically observed scenarios:

Here, we illustrate the framework in the context of empirical data, i.e., with empirically observed values of heritability, disease prevalence, and LEV frequency for traits in the UK Biobank (involving up to 361,194 samples), to (1) demonstrate that the predictions and findings generated based on the models of the framework recapitulate empirical results and (2) show real-world applications of the framework. The common variant-based ($\text{MAF} \geq 0.01$) heritability was obtained from the Ben Neale lab (<http://www.nealelab.is/uk-biobank/>). In total, 2,322 traits were included after removing non-significant ($P_{h^2} > 0.05$) or low-heritability ($h^2 < 0.05$) traits. We leveraged the effect size (OR) and allele frequency for the LEVs, as presented in a recent exome-wide phenome-wide association study in the UK Biobank⁸.

We have also revised the following paragraph describing the use of PB to identify at-risk individuals comparable to LEVs:

Application of PB to identify at-risk individuals comparable to LEVs

For clinical application, it is of considerable interest to determine, from simulations, to what extent one can use the common-variant polygenic burden to identify at-risk individuals with PB comparable to large-effect mutations. We calculated the OR of the LEV (by taking LEV carriers and non-carriers as the exposed and the unexposed group, respectively) and the OR of PB (by taking the top 1%, 5%, and 10% of the PB-ranked samples as the exposed group and the remaining samples as the unexposed group). In general, the ORs of the PB at the various PB cutoffs were highly comparable. For our assumed simulation settings (under a polygenic genetic architecture), the OR of the LEV and OR of the PB showed a similar order of magnitude. The OR of the LEV and that of the PB tended to converge as h_{PB}^2 increased (Figure 7a) or as the heritability due to the LEV decreased (Figure 7b). Assuming h_{PB}^2 to be ≤ 0.3 , we observed that the OR of the LEV was comparable to the OR of the PB (using any of the three PB cutoffs) when h_{LEV}^2 was in the range between 0.01 and 0.03 (Figure 7b). However, as h_{LEV}^2 increased, the two odds ratios would increasingly diverge. Nevertheless, at least based on the estimates of h_{LEV}^2 from available empirical data in a large-scale biobank (UK Biobank), the condition $h_{LEV}^2 > 0.03$ would appear to be rather uncommon (and indeed the maximum h_{LEV}^2 estimate was 0.025 among the 36 heritable traits with an identified LEV, from studies using the UK Biobank^{7,8} [Supplementary Table 3 and Methods]). Thus, our simulations identified a specific range of these parameters wherein the OR of the LEV and that of the PB would be expected to be similar. As expected, lower LEV allele frequency would result in increased OR of the LEV (Figure 7c). The OR of the LEV and that of PB tended to become more similar with higher disease prevalence, i.e., for more common diseases (Figure 7d). Similar patterns were observed for the other two classes of genetic architectures (Supplementary Figure 4 and 5).

10. I find the conclusions much too strong given the results: “our study suggests that cSEV-PB may be a useful addition to the prioritization schemes of samples likely to harbour rare pathogenic variants”. How was this demonstrated? For what diseases and under what architectures?

RESPONSE: We thank the reviewer for this comment. As we mentioned in the responses to #6 and #7, we have applied our summary-statistics-based framework to empirical data from the UK Biobank to demonstrate how the PB may be used as a prioritization scheme, i.e., an LEV screening tool. To illustrate, we determined the probability of an

LEV carrier given case status $P(LEV | Y = 1)$, which we calculated using the 'number of LEV carriers per 1000 cases', for a given PB profile to make predictions in actual empirical datasets with matching parameters used in the simulations. As shown in response #7, the predictions of the framework were highly concordant with the empirical observations from the UK Biobank. We conducted this analysis for four traits, which are available for this comparison⁷: breast cancer, colorectal cancer, type 2 diabetes, and short stature. The framework's parameters (including the common variant based heritability; disease prevalence; allele frequency and odds ratio of the LEV) were estimated from the UK Biobank under the genetic architecture consistent with negative selection¹⁰. Details are included in the response to comment #7 and a new Figure 10.

Reviewer #2 (Remarks to the Author):

The manuscript by Zhou et al. describes simulation studies of the correlation of rare large effect variants and common variant (polygenic) background in disease using different genetic architectures, disease liability models, and study parameters. The simulations provide a quantification of power and utility by varying assumptions of study parameters that are disease and sample specific. They then provide a brief analysis on empirical data on three diseases from recent studies. Overall, the study provides important methodological insight into a clinically important topic that has been subject of multiple recent publications on monogenic-polygenic interaction and disease penetrance across a variety of diseases. The manuscript is well written for the most part, particularly as it details the importance of the subject with regards to clinical translation and could help investigators studying the subject on empirical data for specific disease as they think about the study design and analysis. I hope the below comments could help the authors improve the manuscript prior to publication.

RESPONSE: We thank the reviewer for the very constructive feedback, which we used to substantially improve the manuscript. We thank the reviewer for the positive feedback, i.e., that our framework provides important methodological insight into a clinically important topic. In the revision, we further strengthened the clinical implications of the framework. We applied our summary-statistics-based framework to the UK Biobank and, in addition, confirmed the predictions of the framework with empirical observations. Please see our detailed responses, including the specific changes we have made to the manuscript in response to the reviewer's input.

1- While the background, importance and hypothesis are clearly described upfront, the reader struggles to understand in brief what are the methods. While the different parameters and models used for the simulations are made clear, some important details regarding samples are left from the abstract. On page 5, there is discussion of case-only analysis, but it is not clear how that factors into this study. Also, it is not until page 14 that the 23,294 participants from BioVu are mentioned and it is not clear how those are used. For the empirical analysis on the three diseases, important sample details in supp tables 1-3 should be described in the main manuscript of the text.

RESPONSE: We thank the reviewer for the valuable comments and suggestions. We have now provided important details to better clarify the framework. We now emphasize – in the Abstract – that the framework's parameters were informed by empirical data (Vanderbilt's BioVU repository and the UK Biobank). The real genomic data from 23,294 ancestrally European samples (BioVU) were used for previous analyses. The revised manuscript has added additional analyses of the UK Biobank. For

the additional analyses, the genotype was simulated using PLINK with the sample size matched with the UK Biobank (Supplementary Table 3). We used the distribution of MAF and the LD score as calculated from real genomic data from 23,294 BioVU European ancestry samples (page 20, **if not specified, all the page numbers in our responses refer to the revised version of the manuscript with TRACK CHANGES**). Sample details for the disease datasets used in our study, including empirical datasets in the UK Biobank (n up to 361,194) for 36 heritable traits with at least one LEV, are now summarized in Supplementary Tables 1-3 and cited in Methods in the main manuscript (pages 19-20).

For the utility and power estimation, we compared the PB between LEV carriers and non-carriers, among cases only. We have clarified the 'case-only' analysis in Methods (page 14) and Supplementary Information (page 2 of the revised version of Supplementary Information with track changes).

In the revised manuscript, we include the sample description (including sample size) in the Abstract and Introduction. We also provided more information about how the empirical genetic data were used (see pages 14, 17, and 20).

For the analyses of the three empirical datasets, we included additional details, including the sample sizes (LEV carriers and non-carriers) and how we defined the LEV risk alleles in the main text (see pages 37 and 38).

2- Additionally, the punchline findings or conclusions of the study are not stated upfront and one has to read through to distinguish what is background and rationale and what is actual findings of the study. It would be helpful at the end of the conclusion section to state the main findings of the study in a list format.

RESPONSE: We thank the reviewer for the suggestion. We added the conclusion summarizing our main findings (including below).

In conclusion, we developed a summary-statistics-based framework that utilizes the relationship between PB and LEV among cases for some methodologically- and clinically-relevant applications, including PB-based LEV screening. The framework's testable, falsifiable predictions on the proportion of LEV carriers among cases with a given PB profile under pre-specified simulation parameters were confirmed by empirical data. Application of the framework to a large-scale biobank (UK Biobank) showed that the effectiveness of PB-based LEV screening varied substantially by trait. Taken together, these findings shed critical light on the use of PB in clinical practice.

3- While the bulk of the manuscript is describing simulations, it would be helpful to provide results from empirical analysis on more than the three disease examples, particularly to illustrate whether some of the hypotheses generated based on simulations do actually hold in the context of empirical data for diseases with variable heritability, diseases prevalence and LEV frequency.

RESPONSE: We thank the reviewer for the valuable suggestion. We have now further applied our framework to 36 traits (Supplementary Table 3) with empirical parameters (including heritability, disease prevalence, and LEV frequency) derived from the UK Biobank. The common-variant-based heritability had been derived (Neale Lab) from chip-based common SNPs ($MAF \geq 0.01$). The effect size and MAF of LEV were derived from variant-level Exome-wide association study of the UK Biobank samples⁸. The heritability attributable to the LEV was estimated from the “case status by LEV status” table (if available) or from the OR estimate from logistic regression using an interpretable measure of R^2 (coefficient of determination, see Methods on page 19-21 under the section *Application to large-scale biobank*)⁹. Leveraging these empirically-informed parameters, we applied the summary-statistics-based framework and calculated a metric, i.e., the number of LEV carriers per 1000 cases for each of 10 equally-sized bins of PB, from the simulations to make predictions about LEV screening given a PD profile. The framework would predict that cases from the low-PB bins had a higher proportion of LEV carriers, indicating that we would expect to find more LEV carriers among cases with low common-variant-based genetic risk scores (Figure 9, included below). As an example (Figure 9a), we would expect to find 27 cases carrying a stop-gain mutation ($OR = 5.1$, $MAF = 0.0013$) on *MYOC* per 1000 glaucoma cases from the lowest PB risk bin. However, among the 1000 glaucoma cases in the highest PB risk bin, we would expect to find only 6 carriers. We analyzed four additional traits (Figure 9b, 9c, 9d, and 9e) representing three distinct patterns. The results on malignant melanoma (Figure 9b) indicated that only cases with low PB risk would be worth sequencing for LEV screening. For both Crohn’s disease (Figure 9c) and acute tonsillitis (Figure 9d), the striking difference in the estimated proportion of LEV carriers among cases from the different PB risk bins would suggest the usefulness of PB-based prioritization. Per 1000 Crohn’s disease patients, 214 cases would be expected to be *NOD2* frameshift mutation carriers from the lowest PB risk group whereas only about 67 cases from the highest PB risk group would be expected to be carriers of the mutation (Figure 9c). Nearly half of the acute tonsillitis cases (452 / 1000) from the lowest PB risk group were predicted to be *OXCT2* missense mutation carriers, which was a nearly 3-fold increase relative to the highest PB risk group (Figure 9d). However, PB-based prioritization appears to be less helpful for screening low frequency, large-effect type 2 diabetes-associated variants (Figure 9e). As we showed in Figure 9 and Supplementary Figure 16 (included below), the effectiveness of utilizing PB risk for sequencing prioritization varied by trait. We also provided a new script

([github.com/gamazonlab/Polygenic Background Rare Variant Axis](https://github.com/gamazonlab/Polygenic_Background_Rare_Variant_Axis)) that allows users to apply the summary-statistics-based methodology to their datasets with their custom parameters and models.

We further compared the predictions of the framework with empirical observations in the UK Biobank (for four traits: breast cancer, colorectal cancer, type 2 diabetes, and short stature, which are available for this comparison⁷; see Methods). The framework shows that low-PB patients are more likely to be LEV carriers (Figure 10 [included below]). Furthermore, the framework makes predictions on proportion of LEV carriers for a given PB profile. The predicted results are shown in the boxplot (included below) and are concordant with the empirical data (red diamond) from the UK Biobank.

Figure 9. Cases with low polygenic risk score have higher probability of carrying an LEV. Based on empirically-informed parameter values (i.e., derived from the UK Biobank), we performed simulations (under the liability-threshold model and the genetic architecture in line with negative selection) and compared the number of LEV carriers per 1000 cases among the different polygenic risk scores for (a) glaucoma, (b)

malignant melanoma, (c) Crohn's disease, (d) acute tonsillitis, and (e) type 2 diabetes. For each trait, we grouped the cases into 10 equally-sized polygenic risk score bins. For each bin, the mean \pm 1sd of the number of LEV carriers per 1000 cases is displayed in blue circles and bars. The distribution of polygenic risk score is shown as a histogram. For example, per 1000 Crohn's disease cases, 214 would be expected to be *NOD2* frameshift mutation carriers from the lowest PB risk group; in contrast, PB-based LEV screening would be less effective for type 2 diabetes. The proportion of LEV carriers in the lowest PB risk group for acute tonsillitis was a nearly 3-fold increase relative to the highest PB risk group. Thus, the framework (assuming pre-specified simulation parameters) provides testable predictions on the number of LEV carriers per 1000 cases with a given PB profile and on the sample polygenic risk score profile to optimize LEV screening.

Supplementary Figure 16. Cases with low polygenic risk score have higher probability of carrying an LEV. Based on empirically derived parameters from the UK Biobank, we performed simulations (under the liability-threshold model and the genetic architecture in line with negative selection) and compared the number of LEV carriers per 1000 cases among the different polygenic risk scores for 31 UK Biobank traits (additional to Figure

9). For each trait, we grouped the cases into 10 equally-sized polygenic risk score bins. For each bin, the mean \pm 1sd of the number of LEV carriers per 1000 cases is displayed in blue circles and bars. The distribution of polygenic risk score is shown as a histogram. Thus, the framework (assuming pre-specified simulation parameters) provides testable predictions on the number of LEV carriers per 1000 cases with a given PB profile and on the sample polygenic risk score profile to optimize LEV screening.

Figure 10. Prediction of the framework on proportion of LEV carriers in cases with a given PB profile matches empirical observations, providing an LEV screening approach. Using empirically-informed values (i.e., derived from the UK Biobank) of the parameters (including the common variant based heritability; disease prevalence; allele frequency and odds ratio of the LEV), we performed simulations for four traits under the genetic architecture in line with negative selection. We varied the π_0 (the proportion of non-causal variants in the polygenic risk score) at (a) 0, (b) 0.5, and (c) 0.8. Cases were defined under the liability-threshold model and classified into ‘low-PB’ and ‘high-PB’ (using the median as the cutoff) groups. The proportion of LEV carriers in each group was defined as the number of cases carrying LEV over the total number of cases. The distribution of the proportion (across 500 simulated sets) is shown in the boxplot. The median of the proportion is visualized as a black segment in the middle of the box. The lower and upper hinges correspond to the first and third quartiles (the 25th and 75th percentiles). The upper / lower whisker extends from the hinge to the largest / smallest value no further than / at most $1.5 * \text{IQR}$ from the hinge (where IQR is the inter-quartile range or the distance between the first and third quartiles). The actual observed proportion of LEV carriers for each PB profile from empirical data (with the matching

parameters as the simulations) is marked as a red diamond. Thus, the prediction of the framework and the empirical dataset (with matching parameters as the simulations) were concordant.

4- The authors make the argument for reduced polygenic background effect among LEV carriers vs. non-carriers as a generalizable finding based on one recently published example for breast cancer, but it is still not clear that this is generalizable across diseases and studies. It was not seen in OCD in this study, and more notably it was not seen in other recently published studies other than the one they cite. Eg. Kuchenbaecker et al JNCI 2017, Oetgens et al. Nat Comm 2019 and Fahed et al Nat Comm 2020). Although often limited by statistical power, there was no evidence of interaction between LEV and polygenic background in those studies, and significant stratification of risk by polygenic background among LEV carriers. It is helpful to discuss the simulations in this study in context of some clinically important disease examples such as those recently reported.

RESPONSE: We apologize for the confusion regarding the “interaction” result. In the original Figure 7, the error bars denote the interquartile range of ORs (P25 to P75) which were misleading. We have updated the figure by using the 2.5 to 97.5 percentile range for the OR of PB (per sd change) which covers 95% of the simulated ORs. In general, the 2.5 to 97.5 percentile range among LEV carriers and non-carriers overlapped for each simulation setting (Figure 8, included below). No difference in the OR of PB was observed between LEV carriers and non-carriers (i.e., no evidence of interaction between PB and LEV), **which is consistent with the simulation’s assumed additivity of effects but also with recent empirical studies for some traits**^{4, 5, 6}. However, for a given change in h_{PB}^2 , our framework would predict that change in OR of the PB should be generally higher in non-carriers than carriers (Figure 8a, included below) while a more limited differential change in OR between carriers and non-carriers was observed by varying h_{LEV}^2 , f , and K (Figure 8b, 8c, and 8d). Thus, among the tested parameters, h_{PB}^2 is the most important determinant of how differently, between carriers and non-carriers, the OR of the PB changes.

Although these results assumed additivity (no interaction), we then proceeded to evaluate the PB-LEV correlation among cases under the assumption of interaction. We have added the new analysis results (Methods starting on page 10 and Results starting on page 30) and a new Figure 6b (included below) showing the robustness of the PB-LEV correlation in the presence of interaction (as well as, separately, the second scenario of potentially dependent PB and LEV, Figure 6a).

most important determinant of how differently, between carriers and non-carriers, the OR of the PB changes, as can be seen from the “slope” at each point.

Figure 6. Robustness to independence and additivity of PB and LEV. (a) To test the effect of potentially dependent PB and LEV (in the general population) on the PB-LEV correlation (in cases), we assumed that the LEV was in high LD ($D'=1$) with the top-ranked common variant (i.e., with the largest effect size) and compared the utility. Here, we set $h_{PB}^2 = 0.3$, $h_{LEV}^2 = 0.01$, $f = 0.001$, and prevalence (K) = 0.01, and assumed a polygenic genetic architecture and the liability-threshold model in the simulations. (b) To test the extent to which potential interactions between the PB and LEV could affect the PB-LEV correlation, we assumed that the interaction between the LEV and the common variant with the largest effect size contributed to the variance of trait ($h_{interaction}^2 = 0.02$) and performed simulations. All other parameters were fixed as in (a).

5- The discussion on how a low polygenic background risk can explain reduced penetrance of LEV carriers under the liability threshold model is important. Recent study by Fahed et al Nat Comm 2020 highlighted this concept across three diseases. Authors comment (last paragraph page 24) that their findings contrast this study, but it is not clear how that is the case, as the comment seems to be consistent with this study.

RESPONSE: We agree with the reviewer. We have made this change, now on page 45.
“Consistent with a recent report ...”

6- Minor comment: references 23 and 49 are duplicated

RESPONSE: We thank the reviewer for the careful reading. The preprint (the original reference 49) has been updated to the official published version (same as the original reference 23).

Reviewer #3 (Remarks to the Author):

Zhou and co-workers create an extensive simulation framework, partly based on real-world data (LP-structure, negative selection constants, known large effect variants, etc) and partly on simulated varied inputs of small and large effect heritabilities, frequencies, proportion of non-causal variants as well as a setup of different genetic architecture models.

It is noted that the simulation framework predictions are concordant with several observations that seems to be well-established, even self-evident. These include: that utility decrease with decreased proportion of causal variants, that it decreases with lower HSNP2 both for common and rare variants. However, several less self-evident predictions are made in the model and these are the key novelty of the manuscript.

RESPONSE: We thank the reviewer for the very constructive feedback. Please see our detailed response, including changes we have made to the manuscript in response to the reviewer's concerns.

I have the following comments to the manuscript:

1

The utility metric is a central part of the first two pages of results and figure 3. According to methods, it is defined as "the expected proportion of simulations for which, among cases, a significantly lower ($P < 0.05$) cSEV-PB in LEV carriers than in non-carriers is observed" (from methods). It took me several reads to understand this. That there is less cSEV-PB in LEV carriers than non-carriers. I if understand correct. And I'm still unsure why this is called utility. Utility towards what? I think it would improve the manuscript to be clearer about what it is, and why it is important - also in the results section (right now it only back-refers to methods).

RESPONSE: We thank the reviewer for the comments. We used the term "utility" as we were interested in quantifying the usefulness of the relationship between the common-variant polygenic burden and the LEV burden among cases in probing the genetic architecture of a trait. Being able to detect the relationship in a clinical setting, we feel, is concordant with the usefulness of the relationship towards the many applications of interest, including the possibility of rare pathogenic variant discovery in low polygenic risk score cases/individuals. This was the reason for the use of the term "utility". In the revised manuscript, we clarified it in the Results (included below).

We use the term utility, as the usefulness of the PB in some of the clinically important applications we have in mind (e.g., as a sample prioritization tool for sequencing or for well-powered discovery of a large-effect pathogenic variant) depends on our ability to detect the PB-LEV relationship from a (clinical) sample of causal variants and of individuals.

2

Because this frame-work could form a standard in future rare and common variant combinations, perhaps you should consider a better and simpler acronym than “cSEV-PB”. I have no specific suggestions but found the current acronym very clunky.

RESPONSE: We thank the reviewer for the comment. We simplified the “cSEV-PB” (short for common, small effect variant polygenic burden) to “PB” (for polygenic burden). We have clarified the definition in the main text without sacrificing the accuracy.

3

A number of very concrete testable hypotheses are posed, but very few of them are tested in application to real world data. Principally the real-world testing only briefly describes testing of the polygenic burden in tourettes and obsessive-compulsive disorder, with or without CNVs. The manuscript would benefit from a clear listing of what other hypotheses are recommendations for future studies.

RESPONSE: We thank the reviewer for the suggestions. We have now (much more comprehensively) applied the framework to 36 complex traits with at least one LEV (Supplementary Table 3). We applied our summary-statistics-based framework to trait-specific, empirically-informed parameters (including heritability, effect size, allele frequency, and prevalence) estimated from the UK Biobank. The common-variant-based heritability had been derived (Neale Lab) from chip-based common SNPs (MAF \geq 0.01). The effect size and MAF of LEV were estimated from variant-level Exome-wide association study of the UK Biobank samples⁸. The heritability attributable to the LEV was estimated from the “case status by LEV status” table (if available) or from the OR estimate from logistic regression using an interpretable measure of R^2 (coefficient of determination, see Methods on page 19-21 under the section *Application to large-scale biobank*)⁹. Leveraging these empirical parameters, we calculated a new metric from the simulations, i.e., the number of LEV carriers per 1000 cases for each of 10 equally-sized bins of PB risk score, to make predictions on PB-based LEV screening. The framework would predict that cases from the low-PB risk bins had a higher proportion of LEV carriers, indicating that we would expect to find more LEV carriers among cases with low common-variant-based genetic risk scores (Figure 9, included below). As an example

(Figure 9a), we would expect to find 27 cases carrying a stop-gain mutation (OR = 5.1, MAF = 0.0013) on *MYOC* per 1000 glaucoma cases from the lowest PB risk bin. However, among the 1000 glaucoma cases in the highest PB risk bin, we would expect to find only 6 carriers. We analyzed four additional traits (Figure 9b, 9c, 9d, and 9e) representing three distinct patterns. The results on malignant melanoma (Figure 9b) indicated that only cases with low PB risk would be worth sequencing for LEV screening. For both Crohn's disease (Figure 9c) and acute tonsillitis (Figure 9d), the striking difference in the estimated proportion of LEV carriers among cases from the different PB risk bins would suggest the usefulness of PB-based prioritization. Per 1000 Crohn's disease patients, 214 cases would be expected to be *NOD2* frameshift mutation carriers from the lowest PB risk group whereas only about 67 cases from the highest PB risk group would be expected to be carriers of the mutation (Figure 9c). Nearly half of the acute tonsillitis cases (452 / 1000) from the lowest PB risk group were predicted to be *OXC72* missense mutation carriers, which was a nearly 3-fold increase relative to the highest PB risk group (Figure 9d). However, PB-based prioritization appears to be less helpful for screening low frequency, large-effect type 2 diabetes-associated variants (Figure 9e). As we showed in Figure 9 and Supplementary Figure 16, the effectiveness of utilizing PB risk for sequencing prioritization varied from traits to traits. In addition, we provided a new script ([github.com/gamazonlab/Polygenic Background Rare Variant Axis](https://github.com/gamazonlab/Polygenic_Background_Rare_Variant_Axis)) that allows readers to perform the simulation and quantify the effectiveness given customized parameters and models.

We further compared the predictions of the framework with empirical observations in the UK Biobank (for four traits: breast cancer, colorectal cancer, type 2 diabetes, and short stature, which are available for this comparison⁷; see Methods). The framework shows that low-PB patients are more likely to be LEV carriers (Figure 10 [included below]). Furthermore, the framework makes predictions on proportion of LEV carriers for a given PB profile. The predicted results are shown in the boxplot (included below) and are concordant with the empirical data (red diamond) from the UK Biobank.

Figure 9. Cases with low polygenic risk score have higher probability of carrying an LEV. Based on empirically-informed parameter values (i.e., derived from the UK Biobank), we performed simulations (under the liability-threshold model and the genetic architecture in line with negative selection) and compared the number of LEV carriers per 1000 cases among the different polygenic risk scores for (a) glaucoma, (b)

malignant melanoma, (c) Crohn’s disease, (d) acute tonsillitis, and (e) type 2 diabetes. For each trait, we grouped the cases into 10 equally-sized polygenic risk score bins. For each bin, the mean \pm 1sd of the number of LEV carriers per 1000 cases is displayed in blue circles and bars. The distribution of polygenic risk score is shown as a histogram. For example, per 1000 Crohn’s disease cases, 214 would be expected to be *NOD2* frameshift mutation carriers from the lowest PB risk group; in contrast, PB-based LEV screening would be less effective for type 2 diabetes. The proportion of LEV carriers in the lowest PB risk group for acute tonsillitis was a nearly 3-fold increase relative to the highest PB risk group. Thus, the framework (assuming pre-specified simulation parameters) provides testable predictions on the number of LEV carriers per 1000 cases with a given PB profile and on the sample polygenic risk score profile to optimize LEV screening.

Figure 10. Prediction of the framework on proportion of LEV carriers in cases with a given PB profile matches empirical observations, providing an LEV screening approach. Using empirically-informed values (i.e., derived from the UK Biobank) of the parameters (including the common variant based heritability; disease prevalence; allele frequency and odds ratio of the LEV), we performed simulations for four traits under the genetic architecture in line with negative selection. We varied the π_0 (the proportion of non-causal variants in the polygenic risk score) at (a) 0, (b) 0.5, and (c) 0.8. Cases were defined under the liability-threshold model and classified into ‘low-PB’ and ‘high-PB’ (using the median as the cutoff) groups. The proportion of LEV carriers in each group

was defined as the number of cases carrying LEV over the total number of cases. The distribution of the proportion (across 500 simulated sets) is shown in the boxplot. The median of the proportion is visualized as a black segment in the middle of the box. The lower and upper hinges correspond to the first and third quartiles (the 25th and 75th percentiles). The upper / lower whisker extends from the hinge to the largest / smallest value no further than / at most $1.5 * \text{IQR}$ from the hinge (where IQR is the inter-quartile range or the distance between the first and third quartiles). The actual observed proportion of LEV carriers for each PB profile from empirical data (with the matching parameters as the simulations) is marked as a red diamond. Thus, the prediction of the framework and the empirical dataset (with matching parameters as the simulations) were concordant.

A new limitation paragraph that we have added comments on the need for confirmation of our findings in additional empirical datasets (for future studies). Towards this end, we are making publicly available software (which we call PB-LEV-SCAN, downloadable on GitHub) to enable application of our summary-statistics-based methodology for testing new hypotheses. By plugging in a user’s own set of parameters from his/her study, one would be able to determine what should be expected (expected distribution) and departure from this expectation. In addition, one would be able to identify possible constraints on the parameters, helping to put constraints on genetic architecture. We have added a new paragraph on new hypotheses as recommendation for future studies, as the reviewer suggested:

Several future applications come to mind. First, as we noted above, estimation of disease prevalence using the genetic risk score follows from the framework. For highly heterogeneous traits, the less noisy genetic risk score may lead to improved estimation. Second, the framework raises the possibility of obtaining an estimate of the proportion of non-causal variants (π_0) in the PB. We provide a software implementation, PB-LEV-SCAN, to enable application of the framework to other complex traits and the testing of other hypotheses in future studies.

We also added the highlighted text below to Methods (under “Generative models of disease liability” on page 9):

We note that we can derive the prevalence K as follows:

$$K = P(Y = 1) = \int_0^1 p\varphi(p)dp = E[p]$$

where $\varphi(p)$ is the probability density of p . As the disease risk probability p is a function of the genetic risk score $x = A + R$, the framework enables estimation of disease

prevalence K using the genetic data (which may be less noisy than the phenotype information).

4

In supplementary figure 10 there's discrepancy between the caption text and the simulated constants printed above the figure, e.g. in figure a, the $h2_{LEV}=0.03$, but in caption it says $h2_{rare}=0.3$, $h2_{rare}=0.03$). Also, it seems that only two different combinations are simulated (according to caption).

RESPONSE: We have corrected this typo in the legend of the Supplementary Figure 10. We thank the reviewer for the careful reading.

5

It is not very easy to reproduce the findings from the code in the github repository. There are many external links (summary `<- read.table(paste0('~/../Dropbox/POLYGENIC_LARGE_EFFECT/simulation/results/result_') and un-stated dependencies, e.g. cmd=paste0('/home/zhoud2/tools/plink2/plink2 - bfile /data/c***/z***/projects/prslev/common/common_all --score). I think this could be improved, as I at least could not reproduce the simulations even after much tinkering.`

RESPONSE: We thank the reviewer for the suggestion. We modified the code to make it more user-friendly. The script for our summary-statistics-based framework is available on the GitHub (github.com/gamazonlab/Polygenic_Background_Rare_Variant_Axis). A detailed document describing input options, interpretation of results/intermediate results, and an example are also provided. The new script performs simulations based on the input parameters (heritability, MAF and effect size of LEV, and prevalence) and empirically informed input data (distributions of MAF and the LD score, estimated from ancestrally European samples). The fact that the framework is built on summary statistics will be extremely useful to the scientific community. The tool will allow users to (1) perform simulations under user-defined settings, including disease model, genetic architecture, and all relevant parameters, (2) estimate the utility/power to detect significantly lower PB in LEV carriers than non-carriers among patients, and (3) calculate the proportion of LEV carriers in cases with a given PB profile, enabling PB-based LEV screening.

Reviewer #4 (Remarks to the Author):

In this paper, Zhou et al. presented a simulation study to assess the power to detect a difference in cSEV-PB among cases between LEV carriers and non-carriers. The main clinical implication is that subjects with lower polygenic burden may be at higher risk of a carrier of a large-effect rare variant. The simulations are comprehensive and sound, here are my concerns and suggestions.

RESPONSE: We thank the reviewer for the extremely helpful feedback, which has substantially improved the manuscript. As the reviewer pointed out, the difference in polygenic burden (PB) among cases between large-effect variant (LEV) carriers and non-carriers has an important clinical implication. Please see our detailed response, including changes we have made to the manuscript in response to the reviewer's concerns.

1) A key assumption is that common and rare variants are independent and additive. Could the authors discuss this further, eg possibility that it could be violated? For example, common variants can act as effect modifiers of rare mutations.

RESPONSE: Several recent studies (using empirical data) show that trait-associated common, small effect variants and large-effect variants were independent and their risk burden additive^{4,5,6}. However, as the reviewer noted, it is possible that the independence and additivity do not hold for some complex traits.

Therefore, we performed simulations of two additional scenarios wherein (1) small-effect common variants and LEV are dependent and (2) the interaction between small-effect common variants and LEV also contributes to disease risk. We have added the following new results and a new Figure 6 (included below).

For the first scenario of dependence, we assumed that the LEV was in high LD ($D'=1$) with the common variant with the largest effect size. We performed simulations and compared the utility of the PB-LEV correlation with the utility when PB and LEV were independent. For the simulations, we set $h_{PB}^2 = 0.3$, $h_{LEV}^2 = 0.01$, $f = 0.001$, and prevalence (K) = 0.01, and assumed a polygenic genetic architecture and the liability-threshold model. The values for the utility were similar for the dependent and independent cases as sample size was varied from 500 to 10,000 (Figure 6a).

For the second scenario of interaction, we assumed that the interaction between the LEV and the common variant with the largest effect size contributed to the trait variance ($h_{interaction}^2 = 0.02$). For all other parameters, we used the same settings as above. Here again, the utility values between the disease models with and without interaction were similar (Figure 6b).

Figure 6. Robustness to independence and additivity of PB and LEV. (a) To test the effect of potentially dependent PB and LEV (in the general population) on the PB-LEV correlation (in cases), we assumed that the LEV was in high LD ($D'=1$) with the top-ranked common variant (i.e., with the largest effect size) and compared the utility. Here, we set $h_{PB}^2 = 0.3$, $h_{LEV}^2 = 0.01$, $f = 0.001$, and prevalence (K) = 0.01, and assumed a polygenic genetic architecture and the liability-threshold model in the simulations. (b) To test the extent to which potential interactions between the PB and LEV could affect the PB-LEV correlation, we assumed that the interaction between the LEV and the common variant with the largest effect size contributed to the variance of trait ($h_{interaction}^2 = 0.02$) and performed simulations. All other parameters were fixed as in (a).

2) Are there ways to link up simulation parameters with real gwas/genomics data? Eg using heritability, N_{poly} and distribution of effect size of variants as derived from some real data?

This could be difficult I suppose but I would suggest also briefly discuss it as a limitation (ie difficulty in choosing parameters that fully match actual diseases).

RESPONSE: We thank the reviewer for the valuable comment. We have now applied our summary-statistics-based framework to the UK Biobank using empirically informed parameters (including heritability, effect size and MAF of LEV, and prevalence) estimated from the UK Biobank. The common-variant-based heritability had been

derived (Neale Lab) from chip-based common SNPs ($MAF \geq 0.01$). The effect size and MAF of LEV were estimated from Exome-sequencing⁸. The heritability attributable to the LEV was estimated from the “case status by LEV status” table (if available) or from the OR estimate from logistic regression using an interpretable measure of R^2 (coefficient of determination, see Methods on page 19-21 under the section *Application to large-scale biobank*)⁹. In total, 36 heritable traits, each with an identified LEV, were included (Supplementary Table 3). Leveraging the empirically-informed parameters, we calculated a metric from the simulations, i.e., the number of LEV carriers per 1000 cases for each of 10 equally-sized bins of PB risk score, to make predictions about LEV screening given a PB profile. The framework would predict that low-PB cases had higher probability of being LEV carriers (compared to high-PB cases), indicating that we would expect to find more LEV carriers among cases with low common-variant-based genetic risk scores (Figure 9, included below). As an example (Figure 9a), we would expect to find 27 cases carrying a stop-gain mutation ($OR = 5.1$, $MAF = 0.0013$) on *MYOC* per 1000 glaucoma cases from the lowest PB risk bin. However, among the 1000 glaucoma cases in the highest PB risk bin, we would expect to find only 6 carriers. We analyzed four additional traits (Figure 9b, 9c, 9d, and 9e) representing three distinct patterns. The results on malignant melanoma (Figure 9b) indicated that only cases with low PB risk would be worth sequencing for LEV screening. For both Crohn’s disease (Figure 9c) and acute tonsillitis (Figure 9d), the striking difference in the estimated proportion of LEV carriers among cases from the different PB risk bins would suggest the usefulness of PB-based prioritization. Per 1000 Crohn’s disease patients, 214 cases would be expected to be *NOD2* frameshift mutation carriers from the lowest PB risk group whereas only about 67 cases from the highest PB risk group would be expected to be carriers of the mutation (Figure 9c). Nearly half of the acute tonsillitis cases (452 / 1000) from the lowest PB risk group were predicted to be *OXCT2* missense mutation carriers, which was a nearly 3-fold increase relative to the highest PB risk group (Figure 9d). However, PB-based prioritization appears to be less helpful for screening low frequency, large-effect type 2 diabetes-associated variants (Figure 9e).

In the revision, we further compared the predictions of the framework with empirical observations in the UK Biobank (for four traits: breast cancer, colorectal cancer, type 2 diabetes, and short stature, which are available for this comparison⁷; see Methods). Under the liability-threshold model, the framework would predict that low-PB cases had a higher probability carrying a LEV than high-PB cases (the median of PB was used to define low and high PB groups). The predicted proportion of LEV carriers for a given PB profile is shown in boxplot in Figure 10a (included below). Notably, the empirical data (marked as a red diamond) are concordant with the simulation results. Under the logit model, we observed concordant results for breast cancer, type 2 diabetes, and short stature (Supplementary Figure 15). For colorectal cancer, a higher proportion of LEV carriers was predicted in low-PB cases, in agreement with empirical data; however, the

predicted proportion of LEV carriers in both low-PB and high-PB groups was slightly higher than observed from the empirical data (Supplementary Figure 15).

We added a new “limitation and future application” section, as the reviewer suggested:

The current study has several limitations. The disease models we used (liability-threshold model and logit risk model) are plausible and wide-ranging, but they are not likely to cover all possible scenarios. Although we tested relaxing the assumptions of additivity and interaction of PB and LEV, comprehensive studies of generative models are needed. Diseases are complex, involving heterogeneous causal etiologies for individual variants and implicated biological processes and pathways, which the framework may not accurately model. Although we validated the predictions of the framework in several disease datasets in a large-scale biobank, the generalizability of the framework needs to be confirmed in additional datasets.

Several future applications come to mind. First, as we noted above, estimation of disease prevalence using the genetic risk score follows from the framework. For highly heterogeneous traits, the less noisy genetic risk score may lead to improved estimation. Second, the framework raises the possibility of obtaining an estimate of the proportion of non-causal variants (π_0) in the PB. We provide a software implementation, PB-LEV-SCAN, to enable application of the framework to other complex traits and the testing of other hypotheses in future studies.

Figure 9. Cases with low polygenic risk score have higher probability of carrying an LEV. Based on empirically-informed parameter values (i.e., derived from the UK Biobank), we performed simulations (under the liability-threshold model and the genetic architecture in line with negative selection) and compared the number of LEV carriers per 1000 cases among the different polygenic risk scores for (a) glaucoma, (b)

malignant melanoma, (c) Crohn’s disease, (d) acute tonsillitis, and (e) type 2 diabetes. For each trait, we grouped the cases into 10 equally-sized polygenic risk score bins. For each bin, the mean \pm 1sd of the number of LEV carriers per 1000 cases is displayed in blue circles and bars. The distribution of polygenic risk score is shown as a histogram. For example, per 1000 Crohn’s disease cases, 214 would be expected to be *NOD2* frameshift mutation carriers from the lowest PB risk group; in contrast, PB-based LEV screening would be less effective for type 2 diabetes. The proportion of LEV carriers in the lowest PB risk group for acute tonsillitis was a nearly 3-fold increase relative to the highest PB risk group. Thus, the framework (assuming pre-specified simulation parameters) provides testable predictions on the number of LEV carriers per 1000 cases with a given PB profile and on the sample polygenic risk score profile to optimize LEV screening.

Figure 10. Prediction of the framework on proportion of LEV carriers in cases with a given PB profile matches empirical observations, providing an LEV screening approach. Using empirically-informed values (i.e., derived from the UK Biobank) of the parameters (including the common variant based heritability; disease prevalence; allele frequency and odds ratio of the LEV), we performed simulations for four traits under the genetic architecture in line with negative selection. We varied the π_0 (the proportion of non-causal variants in the polygenic risk score) at (a) 0, (b) 0.5, and (c) 0.8. Cases were defined under the liability-threshold model and classified into ‘low-PB’ and ‘high-PB’ (using the median as the cutoff) groups. The proportion of LEV carriers in each group was defined as the number of cases carrying LEV over the total number of cases. The distribution of the proportion (across 500 simulated sets) is shown in the boxplot. The

median of the proportion is visualized as a black segment in the middle of the box. The lower and upper hinges correspond to the first and third quartiles (the 25th and 75th percentiles). The upper / lower whisker extends from the hinge to the largest / smallest value no further than / at most $1.5 * \text{IQR}$ from the hinge (where IQR is the inter-quartile range or the distance between the first and third quartiles). The actual observed proportion of LEV carriers for each PB profile from empirical data (with the matching parameters as the simulations) is marked as a red diamond. Thus, the prediction of the framework and the empirical dataset (with matching parameters as the simulations) were concordant.

3) the utility metric is quite key to the assessment. I may have missed it, but do authors compared PRS directly between cases and non-cases, like by a t-test or something like that?

'It (Utility) is defined as the expected proportion of simulations for which, among cases, a significantly lower ($P < 0.05$) cSEV-PB in LEV carriers than in non-carriers is observed'. I wonder if we can quantify how much lower the common variant burden is (eg a min. threshold can be set? Or other methods). For instance, 3% lower cSEV-PB may be significant when sample size is large, but may not be clinically relevant.

I am not sure how best to quantify the utility measure, but I feel that just measuring the proportion of (simulation) cases with sig. higher burden is inadequate.

Therefore I also feel it is still hard to say whether cSEV-PB will be clinically useful yet, as patients can still have similar polygenic burden (PB) but how well can we discriminate carriers and non-carriers?

RESPONSE: We thank the reviewer for the valuable comments and suggestions.

We have included the results of the t-test for the comparison of PB between cases and non-cases in Supplementary Table 4. As expected, higher PB was observed for cases than for controls in the simulations.

In the revised manuscript, we kept the utility as a metric for the usefulness of the PB-LEV correlation for some clinically relevant applications. We calculated another metric from the simulations, i.e., the number of LEV carriers per 1000 cases for a given PB profile, applying the framework to 36 heritable traits in the UK Biobank.

We believe this metric is a clinically relevant metric. The metric quantifies PB-based LEV screening, which we then confirmed using empirical observations in the UK Biobank. See also our response to comment #2.

4) The authors have done extensive simulations, however given our limited understanding of the genetics of most diseases, there are a number of important limitations, and I suggest greater caution when proposing the applicability of the finding in clinical practice.

I therefore suggest to provide a more thorough discussions of the limitations of the simulation approach and studies conducted, eg

- Simulations are useful but it is still hard to mimic real disease genetics, eg no, of causal variants, genetic architecture, normal distribution of effect sizes are very hard to verify
- Assumptions of independence (and additivity) between common and rare variants unlikely to be true in many cases
- Remains to be studied whether the findings are generalizable, given varying genetic architecture of diff diseases
- For OCD (or TS), I suppose the sample size is still not very large and the roles of large-effect rare variants are still under study, so one may not be able to show the relationship between PB and LEV

RESPONSE: We thank the reviewer for the comments and suggestions. We agree with the reviewer for all these points. As we noted in the response to comment #2, we have applied the framework to 36 traits from the UK Biobank to support our claims. For the assumptions of independence and additivity, we performed new simulations to show the robustness of our conclusions under these scenarios, but we also agree that independence and additivity are unlikely to be true in many cases. A new “limitations and future applications” section has been added to the Discussions section (included below).

The current study has several limitations. The disease models we used (liability-threshold model and logit model) are plausible and wide-ranging, but they are not likely to cover all possible scenarios. Although we tested relaxing the assumptions of additivity and interaction of PB and LEV, comprehensive studies of generative models are needed. Diseases are complex, involving heterogeneous causal etiologies for individual variants and implicated biological processes and pathways, which the framework may not accurately model. Although we validated the predictions of the framework in several disease datasets in a large-scale biobank, the generalizability of the framework needs to be confirmed in additional datasets.

Several future applications come to mind. First, as we noted above, estimation of disease prevalence using the genetic risk score follows from the framework. For highly

heterogeneous traits, the less noisy genetic risk score may lead to improved estimation. Second, the framework raises the possibility of obtaining an estimate of the proportion of non-causal variants (π_0) in the PB. We provide a software implementation, PB-LEV-SCAN, to enable application of the framework to other complex traits and the testing of other hypotheses in future studies.

Reference

1. Day FR, Loh P-R, Scott RA, Ong KK, Perry JR. A robust example of collider bias in a genetic association study. *The American Journal of Human Genetics* **98**, 392-393 (2016).
2. Aschard H, Vilhjálmsson BJ, Joshi AD, Price AL, Kraft P. Adjusting for heritable covariates can bias effect estimates in genome-wide association studies. *The American Journal of Human Genetics* **96**, 329-339 (2015).
3. Munafò MR, Tilling K, Taylor AE, Evans DM, Davey Smith G. Collider scope: when selection bias can substantially influence observed associations. *Int J Epidemiol* **47**, 226-235 (2017).
4. Fahed AC, *et al.* Polygenic background modifies penetrance of monogenic variants for tier 1 genomic conditions. *Nature communications* **11**, 1-9 (2020).
5. Kuchenbaecker KB, *et al.* Evaluation of polygenic risk scores for breast and ovarian cancer risk prediction in BRCA1 and BRCA2 mutation carriers. *JNCI: Journal of the National Cancer Institute* **109**, (2017).
6. Oetjens M, Kelly M, Sturm A, Martin C, Ledbetter D. Quantifying the polygenic contribution to variable expressivity in eleven rare genetic disorders. *Nature communications* **10**, 1-10 (2019).
7. Lu T, Zhou S, Wu H, Forgetta V, Greenwood CM, Richards JB. Individuals with common diseases but with a low polygenic risk score could be prioritized for rare variant screening. *Genetics in Medicine*, 1-8 (2020).
8. Wang Q, *et al.* Surveying the contribution of rare variants to the genetic architecture of human disease through exome sequencing of 177,882 UK Biobank participants. *bioRxiv*, (2020).
9. Lee SH, Goddard ME, Wray NR, Visscher PM. A better coefficient of determination for genetic profile analysis. *Genetic epidemiology* **36**, 214-224 (2012).
10. Zeng J, *et al.* Signatures of negative selection in the genetic architecture of human complex traits. *Nat Genet* **50**, 746-753 (2018).

REVIEWER COMMENTS

Reviewer #1 (Remarks to the Author):

The revision of the manuscript has addressed all my previous comments. The authors have done an immense amount of additional work. The applications to real data in particular show the potential use of this framework and what the implications mean. While I am not entirely convinced how widespread the clinical utility will be based on these findings, I have now doubt about the utility of the framework which really enable this discussion in the first place.

This is original and interesting work and I strongly recommend publication.

Reviewer #2 (Remarks to the Author):

In the revised version of the manuscript, Zhou et al. did a great job at addressing all the reviewers' comments including mine, with additional analyses and better presentation of the key findings and implications.

Particularly, they:

- 1- Stated the key findings in the abstract and discussion/conclusions
- 2- Explained the methods in a digestible way to broader audience with focus on clinical and research applications
- 3- Performed important additional analyses that showed concordance of the predictions from their framework with real world empirical data from the UK Biobank for several diseases. Figures 9 and 10 are critical components of the revised manuscript.
- 4- Explored appropriately the question of additive vs. non-additive interactions between PB and LEV.

I congratulate the authors on a work well-done and important contribution to the field.

Reviewer #3 (Remarks to the Author):

The authors have addressed my comments

Reviewer #4 (Remarks to the Author):

The authors have performed extra analyses and simulations and have discussed the limitations of their work more thoroughly in this revised version of the manuscript. I believe my questions have been adequately addressed.

REVIEWERS' COMMENTS

Reviewer #1 (Remarks to the Author):

The revision of the manuscript has addressed all my previous comments. The authors have done an immense amount of additional work. The applications to real data in particular show the potential use of this framework and what the implications mean. While I am not entirely convinced how widespread the clinical utility will be based on these findings, I have now doubt about the utility of the framework which really enable this discussion in the first place.

This is original and interesting work and I strongly recommend publication.

RESPONSE: We thank the reviewer for the kind words and the feedback.

Reviewer #2 (Remarks to the Author):

In the revised version of the manuscript, Zhou et al. did a great job at addressing all the reviewers' comments including mine, with additional analyses and better presentation of the key findings and implications.

Particularly, they:

- 1- Stated the key findings in the abstract and discussion/conclusions
- 2- Explained the methods in a digestible way to broader audience with focus on clinical and research applications
- 3- Performed important additional analyses that showed concordance of the predictions from their framework with real world empirical data from the UK Biobank for several diseases. Figures 9 and 10 are critical components of the revised manuscript.
- 4- Explored appropriately the question of additive vs. non-additive interactions between PB and LEV.

I congratulate the authors on a work well-done and important contribution to the field.

RESPONSE: We thank the reviewer for the summary and the feedback.

Reviewer #3 (Remarks to the Author):

The authors have addressed my comments

RESPONSE: Thank you.

Reviewer #4 (Remarks to the Author):

The authors have performed extra analyses and simulations and have discussed the limitations of their work more thoroughly in this revised version of the manuscript.

I believe my questions have been adequately addressed.

RESPONSE: Thank you.